# Organic carbon recycling in subduction zones

Baptiste Debret ®[1] ✉, Pierre Bouilhol ®[2], Hélène Bouquerel[1], Thomas Rigaudier ®[2], Clément Herviou ®[1], Valentin Desmalles[1], Pierre-André Velut[1,3], Bénédicte Ménez ®[1], Stéphane Schwartz[4] & Pierre Cartigny[1]

Abiotic solid organic compounds constitute a ubiquitous byproduct of oceanic lithosphere serpentinization but their evolution and fate during subduction remains largely unexplored. Here, we assess the role of prograde metamorphism in modifying the chemical structure and isotopic signature of both biological and abiotic organic carbon hosted in sediments and serpentinites, respectively. Our findings demonstrate that these two carbon types undergo distinct maturation pathways under increasing pressure and temperature, with abiotic solid organic compounds retaining H-, O- and N-bearing organic functional groups along subduction. Notably, abiotic solid organic compounds are the sole carriers of isotopically light carbon in eclogitic terrains thanks to silicate armoring. Their recycling into the deep mantle therefore provides a plausible source for the extremely light $\delta^{13}C$ signatures observed in some mantle reservoirs, including sub-lithospheric and eclogitic diamonds, and more broadly represents a key factor in generating mantle carbon isotope variability.

The transfer of carbon from Earth's surface to its interior is primarily regulated by the subduction of oceanic lithosphere (i.e., sediment, crust and mantle) into the deep mantle. This process has long been considered to balance mantle degassing, as carbon isotope ($\delta^{13}C$) analyses of mantle-derived materials – such as lithospheric diamonds and mid-ocean ridge basalts (MORBs) – consistently yield $\delta^{13}C$ values around −5 (±3) ‰ over geologic timescales[1–5]. In contrast, $\delta^{13}C$ values in sub-lithospheric and eclogitic diamonds, which are younger than lithospheric diamonds and exhibit strong geochemical ties to subduction[6], are significantly more variable, reaching down to ‑ −28‰[7]. This pronounced variability likely reflects isotopically heterogeneous carbon fluxes into the mantle, governed by both the nature of recycled carbon sources (i.e., organic vs. inorganic) and differences in carbon recycling efficiency across subduction zones[8–10]. These findings point to a direct link between subduction-related processes and the development of deep mantle isotopic heterogeneities, extending into the transition zone and possibly beyond. Similarly, recent advances in high-precision in situ $\delta^{13}C$ analyses of MORB glasses have revealed an

unexpected diversity in $\delta^{13}C$ values, ranging from −12 to −4‰[11], suggesting that the mantle retains more isotopic heterogeneity than previously recognized. Collectively, these observations call for reappraisal of the long-standing paradigms regarding the carbon isotope composition of the mantle.

A key to understand how subduction zones regulate mantle carbon isotopic composition over geological times is to constrain the evolution of $\delta^{13}C$ of both organic and inorganic carbon during prograde metamorphism along subduction zones. The behaviour of carbon isotopes during subduction is often approximated to that of subducting sediments that are dominating the input of total carbon over the altered oceanic-crust (AOC) and serpentinized peridotite[8,9]. In subducting sediments, inorganic carbonates are isotopically heavy ($\delta^{13}C$ ‑ 0‰) and organic carbon is isotopically light ($\delta^{13}C$ ‑ −22‰)[4]. It is therefore proposed that the subduction of organic-rich sediments can regulate the recycling of isotopically light carbon to the deep mantle and therefore relates to the light $\delta^{13}C$ values of sub-lithospheric and eclogitic diamonds[9]. However, this model is very

---

[1]Institut de physique du globe de Paris, Université Paris Cité, CNRS, Paris, France. [2]Université de Lorraine, CNRS, CRPG, Nancy, France. [3]Sorbonne Université, CNRS-INSU, ISTeP, Paris, France. [4]ISTerre, Université Grenoble Alpes, CNRS, IRD, UGE, Grenoble, France. ✉e-mail: debret@ipgp.fr

difficult to reconcile with the fact that the temperature increase associated with prograde metamorphism will tend to erase by diffusion carbon isotopic heterogeneities between inorganic and organic carbon in metasedimentary rocks[12–15]. The preservation of light $\delta^{13}C$ values would therefore require the near-absence of subducted carbonates. While some subduction systems are indeed organic-rich[10], the key question is whether such settings represent a significant and long-term reservoir of isotopically light carbon for the mantle. At the global scale, carbonates still account for ~80% of the total sedimentary carbon subducted worldwide, suggesting that most metasediments are unlikely to provide the extreme light $\delta^{13}C$ signatures observed in the mantle[4]. Furthermore, subducted sediments may detach from the downgoing slab to form buoyant diapirs, hence reducing the chances for C bearing sediments to be deeply recycled[16]. It is thus primordial to not only better evaluate the impact of diffusion processes on the $\delta^{13}C$ values of metasedimentary rocks in subduction zones but also to start considering other reservoir of organic carbon, such as the AOC and serpentinized peridotites. Here, we take advantage of the Western Alps metaophiolitic system (Europe) to characterize the chemical and isotopic evolution of carbon in metasedimentary rocks and associated metaserpentinites during prograde metamorphism. The latter were recently recognized to contain abiotically formed solid organic compounds[17] that dominate the carbon budget of deep-seated lithologies composing the oceanic lithosphere[18–21]. These compounds represent a previously unexplored pathway for organic carbon recycling into the deep mantle in subduction settings.

## Results

### Geological setting

The Western Alps are one of the best locations worldwide to study subduction-related metamorphism and associated mass transfers. Subducted remnants of the Liguro-Piemont oceanic lithosphere record various metamorphic conditions (Fig. 1a, b) that align with typical P-T gradients for mature subductions[22]. These rocks are little retrogressed during alpine collision thanks to a rapid exhumation[23]. This study focuses on the southern Western Alps (Fig. 1a), where subducted remnants record a wide range of pressure–temperature (P–T) conditions, from low-temperature blueschist facies in the Western Queyras to eclogite facies in Monviso (Fig. 1a). In the Western part of the study area, the Liguro-Piemont fragments of the Queyras Schistes Lustrés complex comprise ~10% of metaophiolitic bodies embedded in a metasedimentary-rich environment (i.e., Schistes Lustrés), made by Upper Jurassic to Upper Cretaceous deposits[24]. This metasediment-dominated domain forms a nappe stack interpreted as a deep paleo-accretionary complex[25] subdivided into two tectono-metamorphic units: the Upper Liguro-Piemont unit, which records low- to medium-temperature blueschist facies conditions (320 °C / 1.2 GPa to 400 °C / 1.9 GPa), and the Middle Liguro-Piemont unit, which preserves higher-temperature blueschist transitional to eclogite facies conditions (415–475°C / 1.7–2.2 GPa)[26]. At the easternmost edge of the study area lies the Monviso meta-ophiolite, an eclogite facies mafic–ultramafic massif extending over 30 km along strike at the French–Italian border (Fig. 1a). It is bounded to the west by the Queyras Schistes Lustrés complex and to the east by the eclogite facies Dora Maira continental unit, with both contacts marked by ductile normal faults[27,28]. The Monviso massif is composed primarily of calcschists, pillow lavas, metabasalts, metagabbros, and metaserpentinites, and it records a metamorphic peak at 520–570°C and 2.6–2.7 GPa[29]. It must be noted that P-T constraints for metaserpentinites remain inherently uncertain due to the limited applicability of classical geothermobarometers in ultramafic rocks. In this study, P–T estimates for metaserpentinites are therefore semi-quantitative and are primarily constrained by temperature-dependent phase transitions, while precised P-T estimates rely on associated lithologies (metabasalts, metagabbros and metasediments; see Supplementary Information Figure S1).

### Solid organic compound characterization

Metaserpentinites and their metasedimentary cover were collected from 6 metaophiolites along the Queyras-Monviso transect, namely the Rocher Blanc, Col Peas, Rocca Bianca, Eychassier, Refuge du Viso and Monviso (Fig. 1a). The carbon bearing phases were characterized using both Fourier Transformed Infrared (FTIR) and Raman spectroscopy.

In the metasediments, both organic and carbonate rich horizons are commonly identified from the Queyras Schistes Lustrés complex.

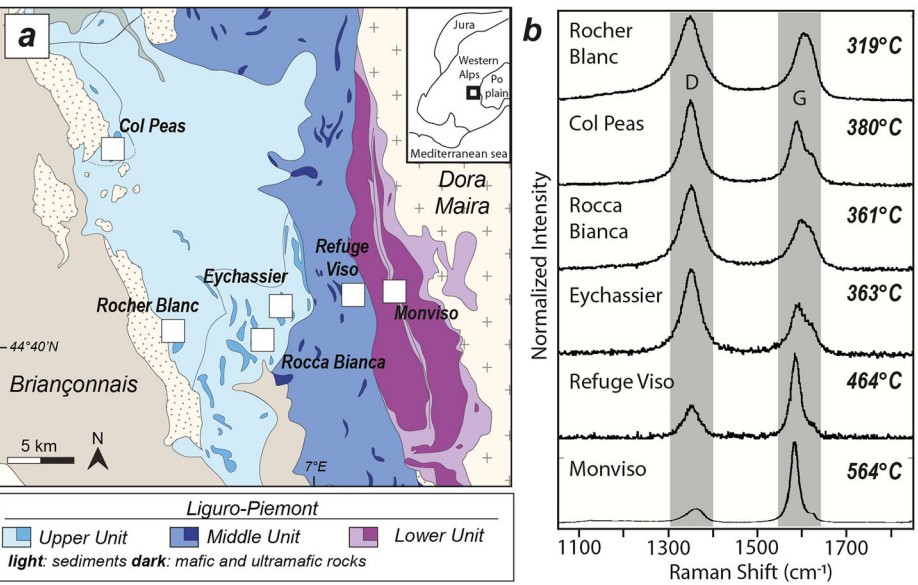

**Fig. 1 | Sampling area and Raman spectra evolution of organic matter with temperature in metasedimentary rocks. a** Geological map of the Queyras-Monviso transect along the Western Alps (modified from[26] with permission from Elsevier). **b** Characteristic evolution of Raman spectra from selected metasediments. Tmax results of organic matter were obtained by the RSCM method (errors are between 30 and 50 °C[68]). Positions of graphite (G) and disorder (D) bands are indicated by grey fields.

Carbonates are mainly calcite/aragonite and ankerite forming large crystals associated with quartz in blueschist and eclogite facies metasediments (Figure S2). Organic carbon forms millimetric, large and elongated dusty aggregates marking the foliation that are associated with quartz, chlorite, phengite, carpholite/chloritoid, lawsonite and garnet. Their Raman spectra can be decomposed in two main bands D (for disorder) and G (for graphite) at about 1350 and 1580 cm⁻¹, respectively (Fig. 1b). The G band is related to the in-plane vibration of carbon atoms in an aromatic ring; this is the only band present in pure graphite. The D band is the result of structural and chemical defects in graphitic carbon. A progressive decrease in the intensity of D band and an increase of G band are observed according to metamorphic grade in blueschist facies units (see also[30]). At eclogite facies P-T conditions, Raman spectra of organic carbon is characterized by an intense and well resolved G band and a small D band (Fig. 1b). The FTIR spectra of eclogitic graphitic carbon are devoid of any significant peaks (Figure S2), in agreement with previous spectroscopic studies of nearly pure graphite[31].

The detailed petrography of the studied metaserpentinites are reported in Schwartz et al.[32] and Caurant et al.[29]. The samples show a progressive metamorphic evolution, whereby lizardite and chrysotile the dominant serpentine species at low-temperature, are replaced by antigorite from low- to medium-temperature blueschist facies P-T conditions. Under high-temperature blueschist facies P-T conditions, antigorite is the sole stable serpentine mineral until the onset of brucite breakdown and the crystallization of secondary olivine and chlorite ± diopside at eclogite facies P-T conditions[32]. The mineral assemblages of metaserpentinites are listed in Table S1. Solid organic compounds form micrometre size dense aggregates of poorly crystalline material filling the porosity of phyllosilicates in the Queyras or Monviso metaophiolites (Fig. 2a, b). These were also observed as ~50 μm wide inclusions within metamorphic olivine in other eclogite facies metaophiolites from the Western Alps (e.g., Zermatt Zaas)[33]. The Raman spectra of the solid organic compounds are characterized by a broad G and D bands centred nearby 1585 cm⁻¹ and 1355–1370 cm⁻¹, respectively (Fig. 2a). Their FTIR spectra show aliphatic C—H stretching bands at 2850 and 2925 cm⁻¹ associated with aromatic C—H stretching bands at 3028, 3059, and 3082 cm⁻¹ (Fig. 2c). Three additional marked bands at 1603, 1493, and 1450 cm⁻¹ can be assigned to aromatic C=C

stretching (± CH₂ scissoring). The band at 1378 cm⁻¹ can be assigned to aromatic C–N stretching. A comparison between antigorite and solid organic compound spectra reveals an additional broad band between 790 and 860 cm⁻¹, which may correspond to N–H or N–O stretching vibrations, but cannot be unambiguously resolved at the micrometric scale. Carbonates were not identified at the microscopic scale.

Focused ion beam (FIB) extraction was realized on one sample from the Monviso massif (Vis20-10) in which abundant solid organic compounds were previously reported[29]. Solid organic material is observed at depth in the sample as filling the space between chlorite sheets (Fig. 3a). The carbonaceous material often embeds nanometric bits of the host mineral (i.e., chlorite). The structure and chemistry of solid organic compounds was characterized by Scattering Scanning Nearfield Optical Microscopy (s-SNOM). Nano-FTIR near-field amplitude spectra reveal nanoscale chemical heterogeneity with two end-member spectral signatures corresponding to chlorite and solid organic compounds (Fig. 3b). The chlorite spectra are characterized by bands at 666, 970 and 1010 cm⁻¹, attributed to Si-O stretching and a large broad band around 1400–1800 cm⁻¹ that can be deconvoluted in three main bands at 1448, 1635 and 1744 cm⁻¹. The solid organic compounds are characterized by additional bands at 845, 894, 1250 and 1466 cm⁻¹. The prominent bands at 845 and 894 cm⁻¹ can be attributed to N-H and/or N-O stretching while the bands at 1250 and 1466 cm⁻¹ can be attributed to aromatic, C-N amines and C = C respectively.

## Carbon bulk rock isotope analyses and nitrogen concentrations

The total carbon concentration [$C_{TC}$] and isotopic composition ($\delta^{13}C_{TC}$), as well as nitrogen [$N$] concentrations analyses were performed using a Thermo Scientific EA IsoLink IRMS System at the CRPG (Nancy, France, see Material and Methods in SI). Metasedimentary rocks are characterized by high $C_{TC}$ ($C_{TC}$ = 0.13–3.27 wt%) and nitrogen ($N$ = 125-463 ppm) concentrations as well as high $\delta^{13}C_{TC}$ (−21.8 to −1.7 ‰) relative to metaserpentinites ($C_{TC}$ = 0.04–0.24 wt%; N = 30-39 ppm; $\delta^{13}C_{TC}$ = −29.9 to −13.0 ‰). These values are similar to previous analyses of Western Alps metasediments and metaserpentinites[12,33–35] (Figure S3a-b). They overlap with abyssal sediments and serpentinites analyses for which $\delta^{13}C_{TC}$ values are primarily controlled by the relative modal abundance of carbonate and organic carbon that preferentially incorporate isotopically heavy and light carbon isotopes, respectively.

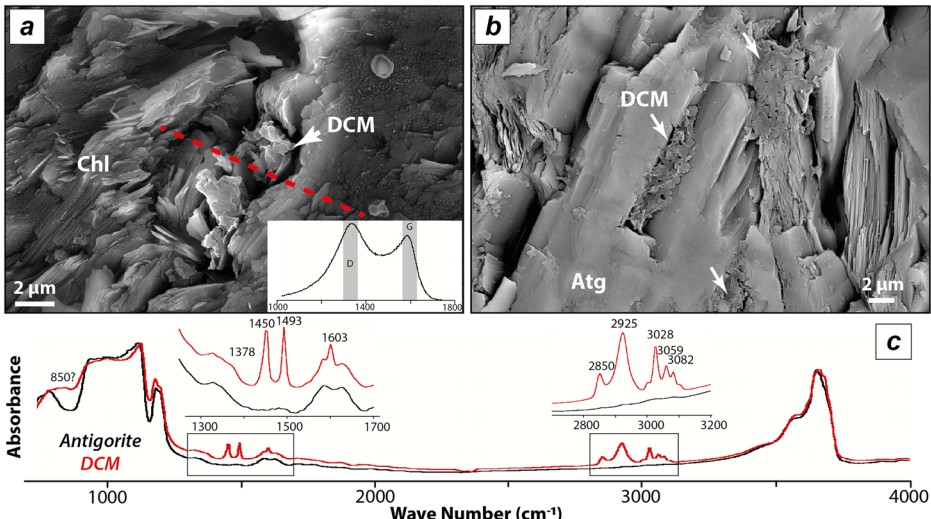

**Fig. 2 | Characterization of solid organic compounds in metaserpentinites from the Monviso massif. a, b** SEM-SE (secondary electron) and -BSE (back scattered electron) images of carbonaceous matter trapped within chlorite and antigorite sheets. The Raman spectra of the solid organic compounds are characterized by large disordered and graphite bands (DCM: disordered carbonaceous matter). The red dotted line indicates the localization of the FIB extraction. **c** Examples of raw FTIR spectra of antigorite and carbonaceous matter associated with antigorite (in red). A focus and annotation are made on organic-related bands. Band assignement are from refs. 69,70. DCM Disordered Carbonaceous Matter, Chl Chlorite, Atg Antigorite.

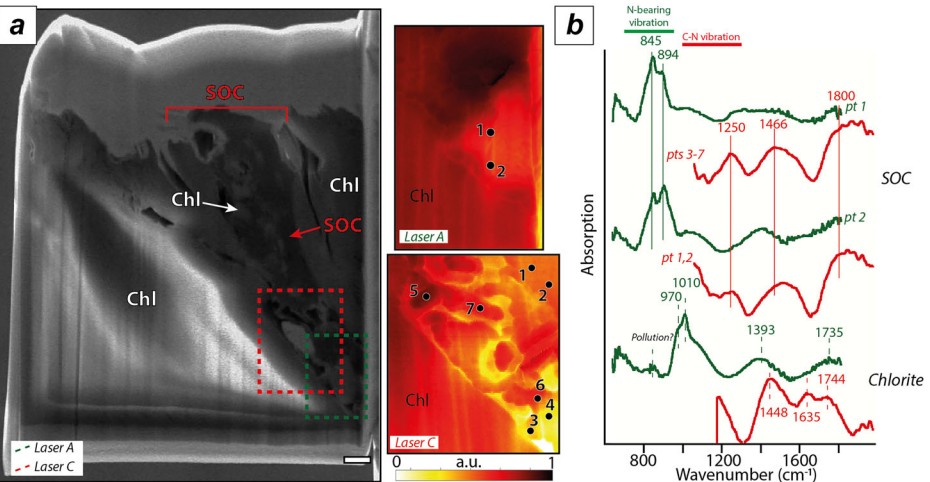

**Fig. 3 | Nano scale characterization of solid organic compound chemistry in metaserpentinites. a** SEM-SE and s-SNOM images of the FIB extraction of solid organic compounds (SOC) filling phyllosilicate sheets. On the SEM-SE image, the dotted rectangles indicate the locations of the repeated s-SNOM images acquired using different broadband laser sources. The s-SNOM analyses are indicated by numbered black dots. The white bar corresponds to 1 μm. a.u., arbitrary units. **b** Nano-FTIR near-field amplitude spectra obtained from the analyses located on (**a**), band assignment is from ref. 69. SOC Solid Organic Compound, Chl Chlorite.

Notably, in abyssal settings, inorganic-rich serpentinites with heavy $\delta^{13}C$ are restricted to surficial alteration zones (< 20–40 m below seafloor, b.s.f.) and low-temperature processes (< 180 °C), whereas organic-rich serpentinites with light $\delta^{13}C$ dominate the C budget of the oceanic lithosphere at greater depth (> 50 m b.s.f.)[19–21,36]. The entire dataset of HP metaserpentinites does not reproduce such a large variability of $\delta^{13}C_{TC}$, with most of the data presenting light $\delta^{13}C_{TC}$ values.

Total organic carbon concentrations [$C_{TOC}$] and isotope compositions ($\delta^{13}C_{TOC}$) were measured on whole-rock sample after HCl fumigation, using the same instruments and calibration protocols as for TC analyses. Because organic carbon can be affected by modern contamination from living bacteria, typically characterized by a *C/N* ratio of ~10[37], potential contamination was assessed by pre-combusting several duplicates of eclogite facies lithologies at 400 °C for 30 min prior to decarbonation, following the protocol of Ader et al.[37]. All precombusted, decarbonated metaserpentinite duplicates yielded results consistent with uncombusted samples (Table S3), indicating negligible modern organic carbon contamination. Total inorganic carbon concentrations [$C_{TIC}$] and isotopes ($\delta^{13}C_{TIC}$) were determined by using an auto sampler Gasbench coupled to a Thermo Scientific MAT 253 continuous flow isotope ratio mass spectrometer (IRMS) at the CRPG (Nancy, France). Total organic (TOC) and inorganic (TIC) carbon concentrations in metasedimentary rocks ($C_{TIC}$ = 0.01–2.52 wt%; $C_{TOC}$ = 0.15–0.76 wt%) are higher than that of metaserpentinites ($C_{TIC}$ = < 0.01–0.06 wt%; $C_{TOC}$ = 0.02–0.16 wt%). Overall, the inorganic carbon dominates the carbon budget of metasedimentary rocks ($C_{TIC}$/$C_{TOC}^{(mean)}$ ~ 6), although large variations exist between graphite and carbonate rich samples ($C_{TIC}$/$C_{TOC}$ = 0.04–10; Figure S3b). Metaserpentinites display a larger proportion of organic carbon relative to inorganic carbon ($C_{TIC}$/$C_{TOC}$ = 0.01–2). The $\delta^{13}C_{TIC}$ of metasedimentary rocks varies in a narrow range of −1.9 to 0.1‰, with the exception of a graphite rich sample (RQ36) having a $\delta^{13}C_{TIC}$ of −8.3 ‰. It contrasts with their highly variable $\delta^{13}C_{TOC}$ ranging from −11.8 to −24.7‰ (Figure S3c, d). The metaserpentinites are characterized by lighter $\delta^{13}C_{TIC}$ and $\delta^{13}C_{TOC}$ relative to metasedimentary rocks. However, a larger variability of $\delta^{13}C_{TIC}$ over $\delta^{13}C_{TOC}$ is observed in the studied samples, with values ranging from −12.3 to −2.1 ‰ and from −36.2 to −28.5‰, respectively.

Interestingly, along the Queyras-Monviso transect, the observed carbon isotope variability ($\Delta^{13}C_{TIC\text{-}TOC} = \delta^{13}C_{TIC} - \delta^{13}C_{TOC}$) in metasediments is intimately correlated with the metamorphic grade, but not for metaserpentinites. In metasediments, while the $\delta^{13}C_{TIC}$ are relatively constant, there is a progressive increase of $\delta^{13}C_{TOC}$ from LT-blueschist ($\delta^{13}C_{TOC}$ = −24.7 to −19.7‰) to HT-blueschist ($\delta^{13}C_{TOC}$ = −21.3 to −13.6 ‰) and eclogitic ($\delta^{13}C_{TOC}$ = −14.1 to −11.8‰) units (Fig. 4b). This agrees with previous studies[12] reporting an overall increase of $\delta^{13}C_{TOC}$, from LT-blueschist ($\delta^{13}C_{TOC}$(Fraitève) = −22.5 ± 0.4 ‰; $\delta^{13}C_{TOC}$(Assietta) = −19.0 ± 1.2‰) to HT-blueschist ($\delta^{13}C_{TOC}$(Albergian) = −17.8 ± 1.8‰) and eclogite ($\delta^{13}C_{TOC}$(Finestre) = −16.6 ± 2 ‰; $\delta^{13}C_{TOC}$(Avic) = −11.8 ± 2.7‰; $\delta^{13}C_{TOC}$(Cigagna) = −16.4 ± 2.3‰) units from other Western Alps localities (Fig. 4b). This geochemical systematic contrasts with the $\delta^{13}C_{TOC}$ of carbonate-poor metaserpentinites that remains constantly light in the different Western Alps metamorphic units, although some $\delta^{13}C_{TOC}$ variations are observed according to localities. In the Queyras Schistes Lustrés complex the $\delta^{13}C_{TOC}$ are constant within an error, from LT- ($\delta^{13}C_{TOC}$(ColPeas) = −28.8 ± 0.4‰; $\delta^{13}C_{TOC}$(RocherBlanc) = −28.5‰; $\delta^{13}C_{TOC}$(RoccaBianca) = −29.6‰; $\delta^{13}C_{TOC}$(Eychassier) = −29.1 ± 0.4‰) to HT- ($\delta^{13}C_{TOC}$(RefugeViso) = −29.6 ± 0.6‰) blueschist units. Larger variations of $\delta^{13}C_{TOC}$ are observed among eclogitic massifs from the Western Alps ($\delta^{13}C_{TOC}$(Monviso) = −33.6 ± 6 ‰; $\delta^{13}C_{TOC}$(Zermatt) = −28.3 ± 1.6 ‰; $\delta^{13}C_{TOC}$(Cima di Gagnone) = −26.4 ± 1.1 ‰). The Western Alps samples display lighter $\delta^{13}C_{TOC}$ than samples from other blueschist and eclogite massifs worldwide ($\delta^{13}C_{TOC}$(Syros) = −26.6 ± 0.6 ‰; $\delta^{13}C_{TOC}$(Almirez) = −25.4 ± 1.7 ‰; Fig. 4a), although few analyses are currently available.

## Discussion

During subduction, kerogens of biological origin in sedimentary rocks transform progressively and irreversibly into pure graphite through pyrolysis. With increasing temperature, heteroatoms are eliminated and the aromatic skeleton grows and rearranges in parallel planes to finally reach the graphite structure[38]. In natural settings, such a process is well characterised by Raman microspectroscopy[30] and illustrated in Fig. 2b. Through a simple visual inspection, the degree of graphitization corresponds to the progressive disappearance of the defect bands (D band), starting at 200 °C, and an increase of the graphite bands (G band) until the formation of pure graphite at 600–700 °C. Graphitization in metasedimentary rocks is poorly sensitive to insensitive to pressure, making it a powerful geothermometer in metamorphic terrains[30]. Estimates of thermal conditions by applying the method of Raman spectroscopy of carbonaceous material (RSCM) on the metasediments are provided in Fig. 1b.

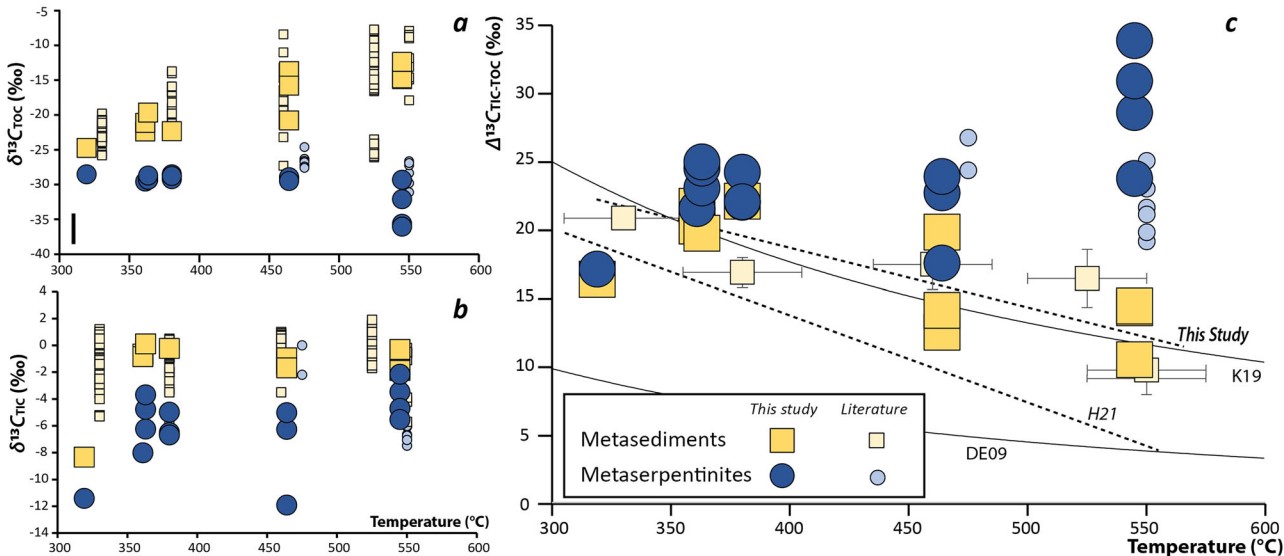

**Fig. 4 | Temperature dependence of carbon isotopic characteristics in metaserpentinites and metasediments. a** $\delta^{13}C_{TOC}$ variations versus temperature estimates. **b** $\delta^{13}C_{TIC}$ variations versus temperature estimates. The black bar represents a 2sd. **c** Evolution of the average carbon isotopic fractionation between inorganic and organic carbon ($\Delta^{13}C_{TIC-TOC}$) in metaophiolites from the Western Alps as a function of temperature. The dataset includes rocks from the Queyras Schiste Lustrès complex (this study and ref. [12]), Syros[71], Cerro del Almirez[34], Zermatt Zaas[33], Monviso (this study and ref. [12]) and Cima Di Gagnone[35] metaophiolites. Literature data are displayed as an average with a 2SE (=2sd/$\sqrt{n}$, $n$ = number of analyses). DE09 and K19 black lines correspond to experimental predictions of carbon isotope fractionations from refs. [40,52]. The dotted lines correspond to empirical fitting of organic and inorganic carbon reequilibration with temperature based on natural data in subduction zones (H21[14]: and this study). Temperature is estimated by RSCM method. TOC Total Organic Carbon, TIC Total Inorganic Carbon.

On Fig. [4]a, the thermal maturation of biotic kerogens in metasedimentary rocks during subduction is associated with a change of $\delta^{13}C_{TOC}$, that increases from about −25‰ in low-temperature blueschist to about −10‰ in eclogite facies units, although local deviation can be observed depending on the initial $C_{TIC}/C_{TOC}$ of metasedimentary rocks[12]. This contrasts with their nearly constant $\delta^{13}C_{TIC}$ through increasing metamorphic grade (Fig. [4]b). Such nearly constant values support only modest decarbonation in alpine metasediments during subduction, as already highlighted by previous isotopic studies[12,39] and flux estimates[9]. Hence, the fractionation of $^{13}C$ between carbonates and organic carbon ($\Delta^{13}C_{TIC-TOC}$) in metasedimentary rocks decreases with temperature (Fig. [4]c). This correlation is commonly interpreted as a reequilibration of carbon isotopes through metamorphic grades following experimental and empirical determination of carbon isotope fractionation between carbonates and graphite that converges toward ~3‰ at 800 °C[40]. Isotopic homogenization between organic and inorganic carbon during prograde metamorphism appears to be a general process at the scale of the Western Alps subduction system (this study, but also[12], see Figure S1a, b for geographical coverage of the dataset in Fig. [4]). Similar behavior is also observed in other paleo-subduction systems: $\Delta^{13}C_{TIC-TOC}$ decreases from 22‰ to 6‰ with increasing temperature in organic-rich eclogites from the Southwestern Tianshan[14], and $\delta^{13}C_{TOC}$ increases with metamorphic grade in metasediments from the Catalina Schists[15]. Altogether, these examples demonstrate that carbon isotope re-equilibration is an intrinsic feature of metamorphism, consistently expressed in subduction settings and already documented in other metamorphic contexts[41]. In abyssal settings, the $\delta^{13}C$ values of organic and inorganic carbon are overall constant through geological times and characterized by values of −22‰ and +1‰, respectively[42]. It therefore appears that the $\delta^{13}C$ of metasedimentary rocks recycled to the deep mantle is mainly constrained by the relative proportion of subducted organic vs inorganic carbon[4]. The budget of organic vs. inorganic carbon in sedimentary rocks varies greatly among worldwide subduction zones, from 0.8% ($C_{TOC}/C_{TC}$ based on flux estimates of organic and total carbon by ref. [10]) in the Philippines to 89% in the Izu-Bonin. Considering a full re-equilibration

during prograde metamorphism between organic and inorganic carbon at the slab scale, this leads to heterogeneous isotopic signatures of metasedimentary rocks that are recycled to the deep mantle in worldwide subduction zones, with values ranging from −19 to +1 ‰ (Fig. [5]a; see Supplementary Information for calculation details). In addition to isotope re-equilibration, carbonate loss through decarbonation or carbonate dissolution may further affect both the carbon budget and isotopic composition of subducting sediments. Recent work by Farsang et al.[43] quantified slab-specific carbonate recycling efficiencies as a function of thermal structure and water availability. To evaluate the isotopic consequences of these processes, we implemented a first-order Rayleigh distillation model using a $CO_2$−calcite fractionation factor of ~3‰ at 700 °C[44] (Figure S4). If present, other aqueous fluid species (e.g., $HCO_3^-$ and $CO_3^{2-}$) mixed with $CO_2$ in slab-derived fluids are expected to reduce this fractionation factor, as they exhibit negative fractionation relative to calcite[45]; the modelled fractionation therefore represents a maximum end-member. The results (Fig. [5]b) indicate that carbonate losses across worldwide subduction systems has a limited impact on the $\delta^{13}C$ signature of recycled sedimentary carbon. In contrast, the efficiency of sedimentary carbon recycling decreases sharply with increasing carbonate loss. Together these findings imply that carbonate mobilization primarily affects the magnitude of carbon transfer rather than its isotopic signature.

To account for the large variations of organic versus inorganic carbon inputs in the subduction zones around the world, we balanced the predicted isotopic signature of metasedimentary rocks to the total of subducted or recycled carbon (Fig. [5]a, b). It appears that the subduction of metasedimentary rocks is unable to recycle a significant amount of isotopically light carbon to the deep mantle with most of the recycled carbon converging toward a narrow range of −10 to +1 ‰ and an average value of ~−8 ‰, identical to that estimated for mantle carbon (−5 ± 3 ‰)[4]. This range is similar to most of mantle derived products, such as MORB, ocean island basalt and lithospheric diamonds from kimberlites[1,5] (Fig. [5]). In these settings, mantle melting and fluids promote the homogenization of mantle carbon isotopic

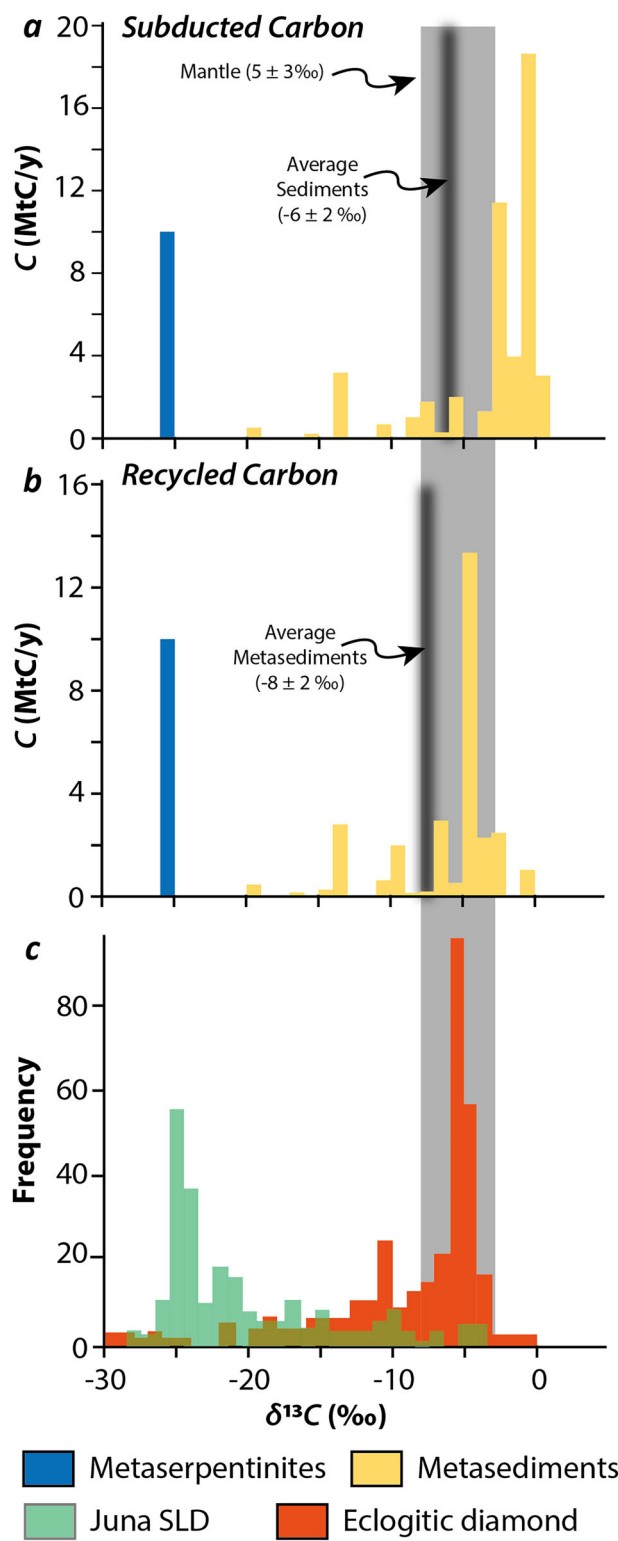

**Fig. 5 | Metaserpentinite and metasediment carbon flux and isotopic composition compared to diamond isotopic composition. The average isotopic composition of worldwide subducted and recycled metasediments, considering inorganic and organic carbon isotope reequilibration, is indicated by black lines and falls within the mantle isotopic range. a** Calculated isotopic composition of subducted metaserpentinites (blue) and metasediments (yellow) vs carbon fluxes in subduction zones (see Supplementary Information for calculation details). The worldwide carbon flux of metaserpentinites and metasediments are from refs. 9,10. Metasediment isotopic composition are calculated from individual subduction zones. **b** Calculated isotopic composition of metaserpentinites and metasediments delivered to the deep mantle accounting for carbonate devolatilization efficiencies from ref. 43. **c** Range of carbon isotopic heterogeneities in sub lithospheric (SLD) and eclogitic diamonds, after[9]. Orange is a compilation of 319 analyses of eclogitic diamonds from ref. 72 and green is the analyses of the Juina SLD from ref. 7. Note that the histogram in (**b**) illustrates the existence of isotopically light carbon in diamonds rather than providing a quantitative distribution of all sub-lithospheric diamonds. The isotopically light cluster highlighted here serves as a qualitative indicator of a distinct recycled carbon component, not as a proxy for global mantle carbon budgets.

The studied metaserpentinites represent abyssal serpentinites that were progressively dehydrated under various P-T conditions during subduction[32]. In metaserpentinites, solid organic compounds preserve a large structural disorder in Raman spectra (Fig. 2a), whatever the metamorphic P-T conditions and the host mineral (i.e., antigorite, chlorite or olivine[33]). The observation of solid organic compounds in HP-HT metamorphic phases (Fig. 2a, b & Fig. 3, see also ref. 33) suggests that these were trapped during the growth of metamorphic minerals. Hence, the thermal maturation of metaserpentinite-hosted solid organic compounds is not compatible with a kerogen-like maturation, calling for other reactional pathways. Besognet et al.[46] recently tested the thermal maturation of synthetic polycyclic aromatic hydrocarbons (PAH; i.e., pyrene, 1-hydroxypyrene, 1-pyrenebutyric acid), considered analogues of abyssal solid organic compounds, at HP-HT conditions (3–7 GPa; 700–1000 °C). These organic compounds are comparable to abiotic solid organic compounds formed during the serpentinization of peridotite in abyssal settings[47]. During their thermal maturation experiments, these compounds preserve a large amount of structural disorder forming hydrogenated (±oxygenated) graphitic carbon at 1000 °C. The Raman spectra reported in solid organic compounds in eclogite facies metaserpentinites (Fig. 2a) are similar to those observed during the thermal maturation of PAH in this experimental work of ref. 46. Indeed, both FTIR and s-SNOM analyses of solid carbonaceous compounds formed at eclogite facies also show the presence of aromatic C-H structures associated with aliphatic C-H$_X$ and N-bearing functional groups (Figs. 2c and 3c), compatible with a hydrogenated (±oxygenated) graphitic structure. It is interesting to note, that not only hydrogen is observed but also prominent N-bearing functional groups suggesting that N remains a major component of the disorder observed in solid organic compounds at HP-HT conditions. The structure of these compounds differs significantly from those formed in open-system HP conditions, such as in forearc serpentinites, where organics are dominated by a strong aliphatic structure[36].

The $\delta^{13}C_{TOC}$ remains constantly light in metaserpentinites from the Queyras-Monviso transect or other HP-HT metaophiolites, ranging from −36 to −28 ‰ with an average value of −31 ± 6 ‰ (Fig. 4a). Such low $\delta^{13}C_{TOC}$ values sharply contrast with the relatively heavy $\delta^{13}C$ signatures (≈ −5 to 0 ‰) reported for metamorphic graphite produced by the reduction of sedimentary carbonates in the Western Alps[48,49]. This distinction supports a fundamentally different origin for the organic carbon preserved in metaserpentinites, consistent with inheritance from abyssal serpentinization rather than in situ carbonate reduction during subduction. Little variations of the $\delta^{13}C_{TIC}$ are

heterogeneities, resulting in an apparent convergence between the isotopic signature of recycled carbon and that of mantle-derived products. However, even if organic and inorganic carbon are not fully homogenized at the scale of the entire mantle, the limited $\delta^{13}C$ variation accompanying inorganic and organic reequilibration in organic rich subduction systems (Fig. 5) is insufficient to explain extremely light values observed in some sub-lithospheric and eclogitic diamonds[4], indicating the contribution of an additional recycled carbon component, or other yet unidentified processes capable of generating large isotopic fractionations.

observed (from −12 to −2 ‰; $\delta^{13}C_{TIC}^{mean}$ = −6 ± 5 ‰; Fig. 4b), this leads to a $\Delta^{13}C_{TIC\text{-}TOC}$ ranging from 15 to 33 (Fig. 4c). These values do not correlate with the temperature record of the metaophiolites and are hence incompatible with a reequilibration of carbon isotopes during prograde metamorphism. Such a disequilibrium could result of either a late organic or inorganic carbon contamination by, respectively, living biomass or $CO_2$-bearing metamorphic fluids, and/or an early armouring of carbon by silicates, hence hindering the isotopic exchange between organic and inorganic carbon during prograde metamorphism. Living bacteria have an atomic $C_{TOC}/N$ of 10 and negative $\delta^{13}C_{TOC}$ (~ −25 ‰[37]). In contrast, the metaserpentinites investigated here exhibit a wide range of $C_{TOC}/N$ ratios, from 7 to 46. Moreover, a biological contamination of the studied samples will necessarily tend to increase $N$ content relative to $C_{TOC}$ leading to positive correlation between $N$ and $C_{TOC}$ contents in metaserpentinites. However, such a correlation is absent in the studied samples, and no relationships between $C_{TOC}/N$ and $\delta^{13}C_{TOC}$ was observed (Figure S5). Importantly, no correlation is observed between $\delta^{13}C_{TOC}$ and indices of external fluid circulation, such as Fluid Mobile Element concentrations (Figure S6), indicating that the organic compounds in these metaserpentinites do not originate from large-scale metasomatic processes. On the other hand, armouring satisfies both microstructural and geochemical observations. Indeed, in contrast to metasediments, where carbon occurs as a mixture of graphite and carbonates (Figure S2) closely associated at the microscale, facilitating isotopic exchange during recrystallisation, metaserpentinites contain almost exclusively solid organic carbon hosted within metamorphic products such as antigorite, chlorite (Fig. 2a, b) or olivine[33], isolating them from the surrounding porosity. These minerals are formed during the early stages of subduction, at temperature ranging from 300 to 450 °C[29,32,50]. Both chlorite and olivine are stable up to 800 °C and beyond, making them robust hosts for the preservation of solid organic compounds with light $\delta^{13}C_{TOC}$ during subduction. Solid organic carbon is nearly pure carbon (~100 wt% $C$), whereas carbonates contain only ~10 wt% C; to accommodate the same absolute amount of carbon, carbonates occupy a much larger mineral volume than organic carbon. This larger volume and surface area in metasediments inherently provide more opportunity for isotopic exchange, whereas the compact nature of organic-carbon–dominated phases in metaserpentinites limits re-equilibration. Combined with the lower total carbon contents in ultramafic rocks (Figure S3), these factors make silicate armoring intrinsically more effective in metaserpentinites than in metasedimentary rocks. Most of the $\delta^{13}C_{TOC}$ in metaserpentinites are indeed comparable to that report for abyssal serpentinites (−31 to −20 ‰[18]). Few extremely light values (i.e., < −32 ‰) might reflect either equilibria with highly reduced fluids (e.g., $CH_4$) at abyssal stage or a partial leaching of organic compounds in $CO_2$ rich fluids prior armouring during the early stages of subduction, both processes leading to a decrease of $\delta^{13}C_{TOC}$. As the studied samples display identical $C_{TOC}$ (614 ± 180 ppm, 2sd/$\sqrt{n}$, $n$ = 16) relative to the average value of abyssal serpentinites (509 ± 117 ppm, $n$ = 180[18]), the first hypothesis is favoured. These observations collectively confirm that the $\delta^{13}C_{TOC}$ in metaserpentinites primarily reflect oceanic processes rather than in situ formation or post-subduction fluid alteration. Overall, both petrographic and geochemical observations indicate that solid organic compounds formed during abyssal serpentinization and were subsequently preserved by encapsulation within metamorphic minerals during prograde subduction-related metamorphism.

The origin of the $\delta^{13}C_{TIC}$ signature in metaserpentinites is less clear as these phases were not observed during the petrographic characterization of the samples. The low concentrations of $C_{TIC}$ could either be inherited from the low inorganic carbon concentrations in deep-seated abyssal serpentinites[18,20] or reflect an early leaching of inorganic-rich lithologies during devolatilization, carbonates being highly soluble in serpentinite-derived fluids relative to organic compounds[33,46,51]. The $\Delta^{13}C_{TIC\text{-}TOC}$ of metaserpentinites correspond to a temperature range of < 300–400 °C, applying either the empirical relationship in metasedimentary rocks or Kueter et al.[52] thermometer (Fig. 4c). Although the lowest $\Delta^{13}C_{TIC\text{-}TOC}$ are coherent with an early silicate armouring of carbonates during prograde metamorphic path, the extreme $\Delta^{13}C_{TIC\text{-}TOC}$ (i.e., > 25) rather call for a disequilibrium between carbonate and organic carbon (through for example devolatilization processes or late carbonate precipitation).

The relative contribution of serpentinites in the deep carbon cycle is not well constrained. This uncertainty arises not only from the degree of serpentinization of the subducting lithosphere, but also from the poorly quantified amount of solid organic compounds that can formed in abyssal serpentinites[18]. Most recent estimates suggest a worldwide total carbon influx from serpentinized peridotites of 1.3–10 MtC/yr[9], but these values are likely underestimated because they do not account for off-axis serpentinization. At eclogite facies P-T conditions, the carbon fluxes of the serpentinized oceanic lithosphere is likely to be dominated by organic carbon where HP metaserpentinites display $C_{TIC}/C_{TOC}$ varying from 0.04 to 2 (Figure S3b). When normalized to subduction length (Table S6), the carbon influx from serpentinized peridotites (~255 Mt C/yr/km) is comparable to, or up to an order of magnitude higher than, that of sediments in organic-rich subduction systems such as Izu-Bonin (42 T/yr/km), Kuril (148 T/yr/km), and NE Japan (410 T/yr/km). While lower than carbonate-rich systems, these fluxes indicate that serpentinite subduction represents an efficient and previously underappreciated pathway for the recycling of isotopically light organic carbon into the deep mantle. In addition, experimental investigations of PAH maturation with temperature and pressure show that these compounds are less soluble than carbonates in serpentinite derived fluids[46] and are hence a robust host for HP-HT organic carbon with light $\delta^{13}C$ values. Metaserpentinites therefore constitute a more plausible source of isotopically light carbon in the mantle[11,53] than metasedimentary rocks, as they systematically preserve organic matter with $\delta^{13}C$ values down to −36 ‰ that remain shielded from isotopic re-equilibration during subduction. Similarly, diamond inclusion systematics indicates that multiple lithologies, including both ultramafic and mafic rocks, contribute to the carbon in sub-lithospheric diamonds[54], yet the light $\delta^{13}C$ signatures observed in some diamonds are consistent with a serpentinite-derived component, supported by B and Fe isotope evidence linking serpentinite subduction to diamond formation in the mantle transition zone[55–57]. It must be noted that AOC may also represent a complementary reservoir of organic carbon acquired during abyssal alteration, as suggested by previous studies[3]. However, the fate of this organic carbon during prograde subduction metamorphism remains poorly constrained, limiting its current integration into global models of deep carbon recycling.

Nitrogen and organic carbon cycles are intimately linked, as the deep nitrogen cycle owes its existence to its fixation into organic matter via either biologic[58] or abiotic[59] processes. In metasedimentary rocks, this process is intimately associated with biochemical cycles leading to the fixation of N in organic matter and its transfer into kerogen and then clay minerals during diagenetic processes and thermal maturation up to greenschist facies P-T conditions[60]. At higher P-T conditions, nitrogen is then fix into micas, the latter having $N$ concentrations up to ~2000 ppm in eclogite facies metasediments[61]. Although large variations of $N$ are expected among (meta-) sedimentary rocks, the $C_{TOC}/N$ ratio remains close to 10 in both abyssal sediments and HP metasediments (Figure S5). Such a ratio is concordant with a biological origin of both organic carbon and nitrogen (see above). The constant $C_{TOC}/N$ between metasediments and their abyssal protoliths further indicates that none of these elements was significantly lost through devolatilization processes at eclogite facies P-T conditions. Such an observation is consistent with previous studies of N isotopes in eclogitic terrains[61]. Considering a constant $C_{TOC}/N$ of 10

and an organic flux of 8 to 12 MtC/yr (considering 20% of organic carbon[9] and total carbon fluxes from refs. 8,10), we re-evaluate the amount of subducted N by metasediments to be 0.8 to 1.2 MtN/yr. This value is similar to the value of 0.76 MtN/yr based on N isotope values[62]. However, recent experimental investigations of nitrogen partitioning during the phengite to K-hollandite transition at 10–12 GPa and 800–1100 °C suggest that between 74 and 57% of the nitrogen is volatilized into metamorphic fluids[63], therefore limiting the role of metasedimentary rocks in the deep nitrogen cycle. Such a process cannot be approached through the study of natural samples which record lower P-T conditions than the K-hollandite reaction. If considered, it lowers the global N flux by metasediments to 0.21–0.52 MtN/yr.

In serpentinized peridotites, abiotic solid organic compounds are formed through redox reactions, such as Fischer-Tropsch like reactions, occurring at high temperature (between 100-450 °C)[36,64]. These reactions can lead to aromatic amino acids formation and N storage into solid organic compounds[59]. The observation of matured N-bearing solid organic compounds in HP-HT samples (Fig. 3) suggests that these are capable to retain N through prograde metamorphism. Furthermore, s-SNOM analyses show that phyllosilicates, such as chlorite (or antigorite), barely display detectable N-bounds (Fig. 2c & Fig. 3c), suggesting that nitrogen is mainly hosted within the organic fraction. It is interesting to note that a striking correlation between $C_{TOC}/N$ and $C_{TOC}$ is observed in HP metaserpentinites (Figure S5). Such a relationship differs from what is observed in (meta-)sediments in which organic carbon has a biological origin. The $C_{TOC}/N$ ratio hence constitutes a robust discriminator between biologic or abiotic organic carbon in metamorphic rocks. Furthermore, as the $C_{TOC}/N$ vs. $C_{TOC}$ correlation is preserved through blueschist and eclogite terrains, it indicates that none of these elements was lost through devolatilization processes. In agreement with this, experimental investigations show that PAH are poorly soluble in metamorphic fluids over a large range of P-T conditions (700–1000 °C and 3–7 GPa)[46] and hence constitute a robust host for nitrogen during subduction. Considering that the average $C_{TOC}$ concentrations of 500 to 1000 ppm in abyssal serpentinites[18] will lead to a $C_{TOC}/N$ ratio of 15 to 30 (Figure S5), we re-evaluate the amount of subducted N in metaserpentinites for an influx of organic carbon of 1.3–10 MtC/yr[9] at 0.04 to 0.67 MtN/yr. Although these estimates remain subject to revision, as the serpentinite reservoir in subduction zones is still poorly constrained, our results overlap with previous nitrogen isotope–based estimates (-0.1 Mt N yr$^{-1}$)[65], but suggest that nitrogen fluxes from serpentinites may reach higher values. It highlights the potential of serpentinites as the main carrier of N during subduction and as a new pathway for the deep nitrogen cycle.

## Methods

### Sample preparation for carbon imaging
Sample preparation for organic carbon characterization was conducted at the Institut de physique du globe de Paris (IPGP, France). Rock samples were cut using sterile ultrapure water to extract the inner core, ensuring the removal of potential post-sampling contamination. The saw was pre-rinsed twice with sterile ultrapure water. The inner core was then handled with clean pliers, thinned, and polished on both sides (down to a thickness of tens of micrometers) using alumina polishing disks with pure ethanol, without the use of resin or glue.

### Fourier-transform infrared microimaging
FTIR microspectroscopy was conducted at IPGP using a Thermo Scientific iN10 MX microscope (Ever-Glo™ conventional infrared source), equipped with a ×15 objective (numerical aperture = 0.7) and a liquid nitrogen-cooled MCT-A detector. The incident beam was collimated to a 20 × 20 μm sample area. The thinned rock sample was deposited on a BaF$_2$ window without any treatment. FTIR hyperspectral maps were acquired in transmission mode over the 4000–675 cm$^{-1}$ range, with a step size of 20 μm, 64 accumulations per spectrum/pixel, and a spectral resolution of 8 cm$^{-1}$. Spectrum analyses and band area-based reconstructions were performed using OMNIC™ software (Thermo Fisher Scientific).

### Raman microspectroscopy
Raman spectra were obtained at IPGP, on resin-free samples with a Renishaw InVia spectrometer using the 514 nm wavelength of a 20 mW argon laser-focused through an Olympus BX61 microscope with a x100 objective (numerical aperture: 0.9, respectively). This configuration yields a planar resolution close to 1 μm. The laser power delivered at the sample surface was 0.5 mW with integration times of 100 s, well below the critical dose of radiation that can damage the carbonaceous matter[66]. Peak position, baseline correction and band widths were determined using PeakFit© software and temperatures were determined using Beyssac et al.[30] methodology, with an uncertainty of about 50 °C.

### Scanning electron microscopy
SEM observations were conducted at IPGP using a Zeiss Auriga FEG-FIB field emission scanning electron microscope. Samples were coated with gold (Au). Images were acquired using a backscattered electron (BSE) detector at high current and a secondary electron (SE) detector at low current, with an accelerating voltage ranging from 10 to 15 kV. Energy-dispersive X-ray spectrometry (EDS) measurements were performed at 15 kV using a Bruker detector.

### Infrared Scattering Scanning Nearfield Optical Microscopy (s-SNOM)
These measurements were conducted at IPGP using an IR neaSCOPE +S Nano FTIR (Attocube), combining atomic force microscopy (AFM) with infrared spectroscopy to achieve nanoscale spatial resolution beyond the diffraction limit and equipped with multiple broadband infrared laser sources: Laser A and B covering 600–1800 cm$^{-1}$ and Laser C covering 1100–1900 cm$^{-1}$. Measurements were conducted in tapping-mode AFM under ambient conditions, using commercial Pt/Ir-coated silicon tips (NanoandMore; Arrow NCPt). The AFM cantilever was driven near its resonance frequency (231–243 kHz) with a tapping amplitude of 105–135 nm. The tip-scattered light is recorded using a Michelson interferometer and demodulated at the second harmonic of the tapping frequency ($n = 2$) to suppress background contributions, providing a spectral resolution better than 7 cm$^{-1}$. Spectra acquisition was performed using Neascan software. To ensure high spectral accuracy and minimize background noise, the system was calibrated using a TGQ1 silicon reference sample. The collected data were post-processed to remove artifacts and enhance signal quality, allowing precise characterization of the chemical composition and vibrational properties of the samples at the nanoscale, using the Neaplot software. This methodology follows established s-SNOM/nano-FTIR protocols, enabling the acquisition of local infrared spectra with lateral resolution on the order of 20–30 nm on biological nanostructures and protein complexes[67].

### Carbon bulk rock analyses
The total carbon (TC) concentrations [$C_{TC}$] and isotopes ($\delta^{13}C_{TC}$) analyses were performed using a Thermo Scientific EA IsoLink IRMS System at the CRPG (Nancy, France). Between 30 to 50 mg of powdered samples were wrapped in tin capsules and then burned at 1020 °C in a combustion reactor consisting of quartz tubes filled with chromium oxide, pure copper and silvered cobalt oxide. Produced gases (N$_2$, CO$_2$) were separated on a chromatographic column maintained at 70 °C and TC concentrations and isotopes were then measured with a Thermo Scientific Delta V Advantage continuous flow isotope ratio mass spectrometer. Total carbon isotope composition was

determined by comparison with CRPG and international standards that were analysed along with samples, the repeated analyses of the standards are (see also Table S2): (i) $\delta^{13}C_{TC\ (BFSd)} = -21.5 \pm 1.1$ (2sd) ‰ ($n = 40$), (ii) $\delta^{13}C_{TC(CRPG\_M2)} = -25.3 \pm 1.8$ ‰ ($n = 24$); (iii) $\delta^{13}C_{TC(IAEA\ 600)} = -27.9 \pm 0.6$ ($n = 13$) ‰; (iv) $\delta^{13}C_{TC(IAEA\ CH6)} = -10.7 \pm 0.6$ ($n = 6$) ‰; (v) $\delta^{13}C_{TC(NBS\ 22)} = -30.2 \pm 0.4$ ‰ ($n = 5$); (v) $\delta^{13}C_{TC(USGS\ 24)} = -16.4 \pm 0.4$ ‰ ($n = 5$); (vi) $\delta^{13}C_{TC(IAEA\ CH7)} = -31.9 \pm 0.02$ ‰ ($n = 2$); (vii) $\delta^{13}C_{TC\ (UB-N)} = -13.8 \pm 0.5$ ‰ ($n = 2$). Values are quoted in the delta notation in ‰ relative to V-PDB. The $2\sigma$ standard error is better than 1.8 ‰ for $\delta^{13}C_{TC}$. Internal standards were used to determine carbon and nitrogen concentrations, the repeated analyses of the standards are: (i) [$C_{TC\ (BFSd)}$] = 0.52 ± 0.04 wt.%,; [$N_{(BFSd)}$] = 580 ± 84 ppm ($n = 40$), (ii) [$C_{TC\ (CRPG\_M2)}$] = 0.40 ± 0.02 wt.%, [$N_{(CRPG\_M2)}$] = 697 ± 61 ppm ($n = 24$); (iii) [$C_{TC\ (EM\_B2153)}$] = 1.91 ± 0.22 wt%, [$N_{(EM\_B2153)}$] = 1351 ± 316 ppm ($n = 8$); (iv) [$C_{TC\ (EM\_B2189)}$] = 0.25 ± 0.02 wt%, [$N_{(EM\_B2189)}$] = 193 ± 51 ppm ($n = 12$); (v) [$C_{TC\ (SOIL\ MIX4)}$] = 2.36 ± 0.01 wt.%,; [$N_{(SOIL\ MIX4)}$] = 484 ± 140 ppm ($n = 11$); (vi) [$C_{TC\ (AVG1)}$] = 0.018 ± 0.010 wt%, [$N_{(AVG1)}$] = 42 ± 12 ppm ($n = 13$); (vii) [$C_{TC\ (UB-N)}$] = 0.063 ± 0.01 wt%, [$N_{(UB-N)}$] = 44 ± 17 ppm ($n = 2$). For [$C$] superior to 0.06 wt% the $2\sigma$ error is better than 15%, while below the $2\sigma$ error decreases down to ~50%. Hence for the [$C$] concentrations below 0.06 wt% was recalculated considering [$C_{TIC}$] and [$C_{TOC}$] (i.e., samples CP8, CE3, CE10, CE14b, RV3 and RV8). The $2\sigma$ error [$N$] is between 5 and 40%. Total organic carbon concentrations [$C_{TOC}$] and isotope compositions ($\delta^{13}C_{TOC}$) were determined for each whole-rock sample using the same instruments and calibration described for analyses of the TC. The inorganic carbon component was removed, prior to each analysis, by HCl fumigation during 5 days at 65°C. The analysis of organic carbon can be affected by modern contamination from living bacteria, which typically have a C/N ratio of ~10[37]. To identify any potential contamination by modern organic matter, six duplicate samples of eclogitic lithologies were pre-combusted off-line in an oven at 400 °C for 30 min prior decarbonation, following the protocol of ref. 37. It should be noted that samples containing organic carbon of lower thermal maturity, such as that found in blueschist terrains, are susceptible to loss N during combustion and were therefore not duplicated. Combusted and uncombusted samples were then run together over a same analytical session. All combusted duplicates of decarbonated metaserpentinites yielded results consistent with uncombusted measurements, within an uncertainty for [$N$] concentration of 3–7% (Table S3), indicating that any contribution from modern organic carbon is negligible relative to analytical uncertainties. Similarly, combusted duplicates of metasediments showed consistent results compared to uncombusted samples, with deviations ranging from 0 to 8%.

Total inorganic carbon isotopes ($\delta^{13}C_{TIC}$) were determined by using an auto sampler Gasbench coupled to a Thermo Scientific MAT 253 continuous flow IRMS at the CRPG (Nancy, France). For each sample, an aliquot of 1 to 100 mg of powder was reacted with 2 mL of supersaturated orthophosphoric acid at 70 °C for 10 hours under a He atmosphere. Values are quoted in the delta notation in ‰ relative to V-PDB. All sample measurements were adjusted to the internal reference calibrated on the international standards, the repeated analyses of the standards are $\delta^{13}C_{TIC\ (BR\ 516)} = -1.2 \pm 0.1$ ‰ ($n = 11$); (ii) $\delta^{13}C_{TIC\ (BR8107)} = -1.1 \pm 0.1$ ‰ ($n = 11$); (iii) $\delta^{13}C_{TIC\ (CA10-08)} = -1.8 \pm 0.1$ ‰ ($n = 2$); (iv) $\delta^{13}C_{TIC\ (NAG7RT)} = -1.1 \pm 0.1$ ‰ ($n = 2$); (v) $\delta^{13}C_{TIC\ (CAL-S\_2003)} = 2.7 \pm 0.1$‰ ($n = 8$). The $2\sigma$ errors was estimated at 1.6 ‰ through repeated analyses of UB-N ($-7.0$‰ $\pm 1.6$‰, $n = 12$). Total inorganic carbon contents [$C_{TIC}$] of the samples were determined by comparison with 4 internal standards consisting in fine-grained marine sediments from the Bay of Bengal and routinely included during the analysis. The repeated analyses of the standards are display in Table S4. The $2\sigma$ errors was estimated to be lower than 35% based on repeated analyses of UB-N ([$C_{TIC}$] = 272 ± 95 ppm, $n = 12$).

### Predicted isotopic composition of metasediments subducted or recycled into the deep mantle

The isotopic composition of metasediments from individual slabs was calculated using a binary mixture between organic and inorganic carbon where $\delta^{13}C = C_{TOC}/C_{TC} * \delta^{13}C_{TOC} + (1 - C_{TOC}/C_{TC}) * \delta^{13}C_{TIC}$. The $C_{TOC}/C_{TC}$ ratios were taken from Table S6. For subducted carbon, constant values of $-22$‰ and $+1$‰ were assumed for $\delta^{13}C_{TOC}$ and $\delta^{13}C_{TIC}$ respectively, while for recycled carbon model also account for slab devolatilization effects, the $\delta^{13}C_{TIC}$ was hence calculated following a Rayleigh distillation model (Figure S4).

## Data availability

The authors declare that the data supporting the findings of this study are available within the supplementary information.

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

## Acknowledgements

This study was supported by the Agence Nationale de la Recherche (ANR) project CARBioNic (ANR-22-CE49-0001-01) awarded to BD. Part of this work was also supported by the IPGP multidisciplinary program PARI, by Region Ile de-France SESAME grants no. 12015908 and EXO47016, and by the IMPACT Project EPHemeris of Initiative d'Excellence Lorraine, a France 2030 program (ANR-15-IDEX-04-LUE) awarded to PB. This study contributes to the IdEx Universite de Paris ANR-18-IDEX-0001.

## Author contributions

B.D. and P.B. designed research, B.D., H.B., T.R, C.H., V.D., and P-A.V. performed research, H.B., T.R., and B.M. contributed analytic tools, B.D., P.B., H.B., T.R., S.S., and P.C. analyzed data, B.D., P.B., T.R, C.H., B.M., S.S., and P.C wrote the paper.

## Competing interests

The authors declare no competing interests.
