## [Transparent Peer Review file · Nature Communications]

Organic carbon recycling in subduction zones

Corresponding Author: Dr Baptiste Debret

This file contains all reviewer reports in order by version, followed by all author rebuttals in order by version

Version 0:

Reviewer comments:

Reviewer #1

(Remarks to the Author)

This manuscript presents an important and timely investigation into how high P–T metamorphism modifies the chemical and isotopic evolution of biological versus abiotic organic carbon in subducting lithologies, including sediments and serpentinites. The study demonstrates that these two carbon types follow distinct maturation trajectories with increasing pressure and temperature. Building on these findings, the authors propose that the recycling of abiotic organic carbon—rather than biological organic matter—provides a plausible source for the extremely light ^{13}C signatures observed in certain mantle reservoirs. This is a novel and thought-provoking hypothesis with potentially significant implications for deep carbon cycling. The manuscript is clearly written, well structured, and supported by solid petrographic and geochemical observations. I recommend acceptance after minor revision.

That said, one methodological issue requires clarification: the pressure–temperature constraints for abiotic organic carbon in the metaserpentinites (Figure 4). Because P–T conditions are intrinsically difficult to determine in meta-ultramafic rocks, the authors should provide a more detailed justification of their approach, assumptions, and uncertainties. Strengthening this point will improve the robustness of the conclusions.

Additionally, I encourage the authors to compare their organic carbon data with metamorphic graphite produced by the reduction of sedimentary carbonates by serpentinite-derived fluids in the Alps. This comparison could further contextualize the isotopic signatures and deepen the discussion regarding the origins of light ^{13}C values.

Below, I list minor comments aimed at further improving clarity and completeness:

Minor comments

1. Lines 65–66:

In addition to the authors' point, it may be worth noting that sediments are often not recycled into the sub-arc mantle due to their relatively low density, which can promote diapiric ascent instead of continued subduction.

2. Line 74:

The term *en route* is French. For stylistic consistency, consider replacing it with an English equivalent.

3. Line 94:

The P–T range of 415–475 °C / 1.7–2.2 GPa corresponds to eclogite-facies conditions. Please clarify this categorization or adjust the phrasing.

4. Lines 133–139:

Please provide appropriate references for the spectral peak assignments.

5. Lines 149–155:

References are also required here. Alternatively, a supplementary table summarizing all peak assignments would improve clarity.

6. Lines 273–274:

The conclusion that the degree of disorder in abiotic organic compounds is not P–T dependent (based on PAH simulations) may require additional high-pressure experimental evidence. Please clarify or temper this statement.

7. Line 285:

Further explanation is needed regarding how the authors confirm that the metaserpentinites reached eclogite-facies conditions, given the well-known difficulty of constraining P–T in meta-ultramafic rocks.

8. Line 339:

Could the generally low temperatures (<300–400 °C) documented for most metaserpentinites be the primary factor driving their low ^{13}C signatures? A brief discussion would strengthen the interpretation.

Reviewer #2

(Remarks to the Author)

Comments to manuscript by Debret et al:

Debret et al., measured the inorganic and organic carbon contents and carbon isotope ratios, together with nitrogen contents, of both meta-sediments and meta-serpentinites with different metamorphic grades, aiming to understand (and emphasize) the role of meta-serpentinites in transferring isotopically light ^{12}C to the mantle, contributing to the diamond-forming fluids with low $\delta^{13}\text{C}$ values (e.g., eclogitic diamonds). They observed the equilibration of carbon isotopes between inorganic carbonate and organic carbon in meta-serpentinites during prograde metamorphism, thus tending to homogenize the $\delta^{13}\text{C}$ values of the sedimentary component. This is however not observed in the adjacent meta-serpentinites, highlighting the role of meta-serpentinites in transporting ^{12}C -enriched organic carbon to the mantle.

Overall, I agree with the proposed idea of meta-serpentinites in transferring ^{12}C -enriched carbon to the mantle, but their relative importance compared to the seafloor sediments and altered oceanic crust needs to be further evaluated. Given the overall quality of the data and the logic that authors applied to reach conclusions, I would recommend major revision after these two main points and the following minor points were reworked.

One of the major comments is: the authors stated that the $\delta^{13}\text{C}$ values of the bulk $\delta^{13}\text{C}$ value of subducting sediments after re-equilibration during prograde metamorphism are comparable to that of the general mantle, thus concluding that sedimentary components cannot transport carbon with low $\delta^{13}\text{C}$ values. This needs to be evaluated, since sedimentary carbonate will also be mobilized out of the slab and the remaining carbonate may not significantly modify the $\delta^{13}\text{C}$ values of the organic carbon. Though this is not observed in your samples, but this loss process of carbonate via decarbonation/carbonate dissolution (etc.) needs to be really discussed, since there are numerous studies discussing the carbonate mobilization from the slab to the forearc and arc volcanoes in modern subduction zones. The authors calculated the bulk $\delta^{13}\text{C}$ values of the subducting sediments of modern subduction zones, so these processes need to be evaluated and is important to your following discussions of the role of meta-serpentinites in deep carbon cycling. If you consider these processes, would the sedimentary component play a role in transferring ^{12}C -enriched organic carbon to the mantle? You may leverage the findings from Farsang et al., 2021 (Nature Communications, Deep carbon cycle constrained by carbonate solubility), where the mobilization of carbonate from slab at different depths have been evaluated (see my detailed comments below). This will need some work in the mass balance calculations of each individual subduction zones to calculate the final $\delta^{13}\text{C}$ values of sedimentary carbonate after some carbonate is lost.

Another major comments: when the authors are trying to emphasize the role of meta-serpentinites in transporting organic carbon to subduction zones, the authors compared the global carbon flux of meta-serpentinites with some individual subduction zones, showing that this global carbon flux of meta-serpentinites are comparable to some subduction zones with sedimentary carbonate-rich lithologies, and a hundred times higher than the subduction zones dominated by sedimentary organic carbon. This does not sound to compare a global flux with an individual subduction zone. For your purpose, the authors should compare unit flux in Mt C/yr/Km to better get a sense of their relative importance. If you do this unit flux comparison, is the meta-serpentinites still important in transporting isotopically depleted carbon to the mantle?

Other comments-Keyed to line numbers:

L 55-56: references for $\delta^{13}\text{C}$ values of organic carbon and carbonate in subducting sediments

L 55: organic carbon is isotopically light

L 126: the mineral assemblage of meta-serpentinite is listed in Table S1.

L 191: lighter

L 223: poorly sensitive to insensitive

L 231-233: Are there any systematic correlations between TIC or TOC contents with metamorphic grades? This could be useful for your argument here to argue against significant decarbonation.

L 234: the average C isotope fractionation between carbonates and solid organic compounds..... Please also report the values of this $\Delta^{13}\text{C}$ based on your samples.

L 252: Please specify here clearer: what is the meaning of full re-equilibration? Do you mean re-equilibration at a certain temperature between inorganic and organic carbon? Or this re-equilibration temperature does not matter, since organic carbon and inorganic carbon will reach equilibrium finally via isotope exchange their overall ^{13}C values will remain unchanged if it is a close system without carbon loss. This is important to your calculation of re-equilibrated $\delta^{13}\text{C}$ values of the subducting meta-sediments at global scale, so this needs to be clearly specified.

L 250-252: The ratio of CTOC/CTC you defined here is the flux of organic carbon divided by total carbon flux (organic + inorganic), right? Please make it clear.

L 516: Pay attention to your mass balance calculation equation: the second item should be $(1-\text{CTOC}/\text{CTC}) \cdot \delta^{13}\text{C}_{\text{TIC}}$? I believe that your calculation results are right (checked), but you just put the bracket in wrong position.

L 262: remove cratons, since kimberlites erupted on cratons.

L 250-270: I agree with your arguments that isotope re-equilibration between organic and inorganic carbon tends to move the ^{13}C value of the entire subducting sedimentary component to higher values and thus erase the $\delta^{13}\text{C}$ values of primary organic carbon with low $\delta^{13}\text{C}$ values of -22 ‰ via equilibration. The final $\delta^{13}\text{C}$ values of the subducting sedimentary component is dependent on the budget of organic vs. inorganic carbon. What if some sedimentary carbonate is removed from the sediments via forearc or arc volcanoes, via decarbonation and carbonate dissolution? Your study is based on Alps samples, and it seems minor decarbonation processes happen, but these effects have been widely observed and modelled in subduction zone settings and are important pathway for carbon removal from subducting slab (see studies on geochemistry of volcanic gases). If organic carbon has remained in the slab as refractory graphite, while some carbonate is removed from the subducting slab. Will this change the overall $\delta^{13}\text{C}$ value of the subducting sediments that are transferred to the mantle? This might not be easy to quantify from a global perspective, but studies from Central America and Izu-Bonin-Mariana have numerous datasets on the fluxes of volcanic CO_2 that is ultimately sourced from the subducting sediments. This needs some discussion and evaluation on your calculation of final ^{13}C values of the subducting sediments to the mantle from a global perspective.

L 275-276: do you mean that these organic compounds were produced and trapped during the fluid-rock interactions inside

the subduction zones? In L 281-282: the authors stated that the organic compounds found in the meta-serpentinite are similar to those found in the serpentinized peridotites in abyssal settings. Is it possible that these organic compounds found in meta-serpentinites of this study come from the seafloor serpentinization processes?

L 295-297, L 197-198: note that in L 197-198, the authors stated that the $\delta^{13}\text{C}$ values of organic carbon and inorganic carbon is larger in meta serpentinites (-36 to -28‰; -12 to -2‰), but in L 295-297, the authors stated that $\delta^{13}\text{C}$ of organic carbon remains constantly (-36 to -22‰), while $\delta^{13}\text{C}$ of inorganic carbon show little variation (-8 to 0‰). The reported $\delta^{13}\text{C}$ values are not consistent between L 197-198 and L 295-297. This needs to be clarified.

Table S5: add the lithologies for each of the sample you have analyzed. It's not very convenient to look at the ID in table S5 and find the corresponding lithology in table s1.

L 304-305: report your CTOC/N ratio of your meta-serpentinite samples here.

L 308-309: is there any correlation between fluid-mobile elements (e.g., Cs, As and B) and TOC or N contents in these meta-serpentinites?

L 314-316 and L 332-333: In L 314-316, the authors stated that the organic carbon was hosted in metamorphic products, such as antigorite, chlorite and olivine, which was formed during early stage of subduction. In L 332-333, the authors stated that the organic carbon was likely sourced from oceanic processes. I am not very clear about these two statements. Do the authors try to argue for/against the origin of organic compounds from serpentinization in abyssal settings or inside the subduction zones? From fluid-mobile elements, it is clear to me that the authors are trying to argue for the production of organic compounds during the oceanic serpentinization processes? This needs to be clarified here.

334-344: carbonate precipitated in oceanic lithosphere could be highly variable and this could be caused by the relative contributions of inorganic carbonate from abiotic processes (general precipitation of bicarbonate from hydrothermal fluids) and microbial processes (see Furnes et al., 2001 and series papers published by him and his colleagues). Microbial fractionation of carbon isotopes in altered basaltic glass from the Atlantic Ocean, Lau Basin and Costa Rica Rift). Then, the obvious question is would you expect that these $\delta^{13}\text{C}_{\text{TIC}}$ are a result of microbial processes?

L 336-338: The highest disequilibrium occurs in the samples with highest metamorphic grade based on Fig. 4c. If it is the leaching (dissolution) of inorganic carbonate during subduction processes, do you see the loss of inorganic carbon during the progressive subduction of the meta-serpentinites based on your samples, particularly those with highest $\Delta^{13}\text{C}_{\text{TIC-TOC}}$, likely representing the most obvious disequilibrium caused by carbonate leaching.

L 345-358: Here, the authors stated that meta-serpentinites could be an important reservoir contributing organic carbon to the mantle sources, which I agree. But they stated that carbon fluxes of global serpentinites are comparable to some subduction zones with carbonate-rich lithologies and extremely higher than the subduction zones dominated by organic-rich sediments. This comparison is not clear to me. It does not sound if you compare a global total flux of carbon carried by serpentinites with a specific subduction zone at a local scale. What if this carbonate-rich/organic-rich sediments dominate the subduction zones at global scale? Then, does these serpentinites still play an important role with comparable carbon fluxes to the sediments? If you really want to emphasize the role of meta-serpentinites in transferring carbon to subduction zones, it is better to compare them with a unit flux, i.e., Mt C/yr/km. This will give a more straightforward to compare between carbon fluxes from meta-serpentinites and sediments. E.g., you can calculate the unit flux of meta-serpentinites from a global perspective (the flux has been estimated by early studies) and compare this unit flux with individual subduction zones (e.g., carbonate-rich versus organic carbon rich). Based on this kind of comparison, do you still see that unit flux of meta-serpentinites is comparable or higher than the sediments carbon flux? Another thought is that: if you want to highlight the subduction or isotopically light carbon to the mantle via subduction of meta-serpentinites, you may simply compare the meta-serpentinites with subduction zones dominated by the subduction or organic carbon, since these subductions are expected to be the main carrier of ^{13}C -depleted organic carbon, compared to the other subduction zones dominated by isotopically heavy carbonate.

L 382-384: when you calculate the flux of nitrogen using CTOC/N, what is the flux of organic carbon did you use for the calculation? This needs to be clearly stated.

L 396-397: there is no detectable N-bonds in chlorite. Did you also test the serpentinite minerals? Do they show any signature of N bonding?

L 410: previous estimation of N flux by seafloor serpentinites are within the range of your estimation of N flux, but your estimation goes to higher values. Make this cleaner.

Other minor comments worth considering:

This manuscript basically compared the relative importance of carbon in meta-serpentinites versus the sedimentary carbon. There are numerous works emphasizing the role of carbon transportation to the subduction zones by altered oceanic crust (AOC). I think it is at least worth some discussion when you emphasize the role of meta-serpentinites in the deep carbon recycling, instead of stating that the carbon fluxes of subduction zones are dominated by sedimentary reservoir and AOC-bearing carbon is not important or negligible (see Alt et al., 1999, GCA, The uptake of carbon during alteration of ocean crust; Coogan and Gills, 2011, Secular variation in carbon uptake into the ocean crust). Instead, in the subduction zones where sedimentary carbon is dominated by organic carbon, AOC carbonate could dominate the budget of subducting slab (such as Izu-Bonin, Tonga, and Kurile; see Farsang et al., 2021; Nature Communication, Deep carbon cycle constrained by carbonate solubility), and in particular, the carbon isotope compositions of carbonate in oceanic crust is highly variable and can also contribute carbon isotopes with extremely low $\delta^{13}\text{C}$ values (see Furnes et al., 2001; Li et al., 2019). I understand that a global perspective of ^{13}C values of AOC is still loosely constrained, but this needs to be mentioned instead of just being neglected.

The authors analyzed the carbon content and isotope compositions, as well as nitrogen contents of meta-sediments and meta-serpentinites by isotope ratio mass spectrometers. I wonder why the nitrogen isotope compositions were not measured, but just nitrogen contents?

Reviewer comments:

Reviewer #1

(Remarks to the Author)

Reviewer #2

(Remarks to the Author)

I have now thoroughly reviewed the revised manuscript. My suggestions have been worked in and the manuscript presents itself significantly improved over last version.

I only have one comment need to be cleared:

Regarding the calculation of isotope fractionation during carbon removal from subducting slab, if I am understanding the model right, the authors considered the decarbonation mechanism for carbon removal as CO₂. How about dissolution of carbonate as bicarbonate in fluids? This will result in different isotope fractionation factors between Calcite-CO₂ and Calcite-HCO₃⁻ (in aqueous fluids) The authors could use carbonate dissolution as another extreme endmember of carbon removal to evaluate its effect on δ¹³C values of the remaining carbon in slab.

After this comment is addressed, I can recommend the revised manuscript for publication in Nature Communications.

REVIEWER COMMENTS

Reviewer #1 (Remarks to the Author):

This manuscript presents an important and timely investigation into how high P–T metamorphism modifies the chemical and isotopic evolution of biological versus abiotic organic carbon in subducting lithologies, including sediments and serpentinites. The study demonstrates that these two carbon types follow distinct maturation trajectories with increasing pressure and temperature. Building on these findings, the authors propose that the recycling of abiotic organic carbon—rather than biological organic matter—provides a plausible source for the extremely light ^{13}C signatures observed in certain mantle reservoirs. This is a novel and thought-provoking hypothesis with potentially significant implications for deep carbon cycling. The manuscript is clearly written, well structured, and supported by solid petrographic and geochemical observations. I recommend acceptance after minor revision.

That said, one methodological issue requires clarification: the pressure–temperature constraints for abiotic organic carbon in the metaserpentinites (Figure 4). Because P–T conditions are intrinsically difficult to determine in meta-ultramafic rocks, the authors should provide a more detailed justification of their approach, assumptions, and uncertainties. Strengthening this point will improve the robustness of the conclusions.

We agree that pressure–temperature constraints in metaserpentinites are intrinsically difficult to quantify and require careful justification. We have now clarified in the manuscript that P–T estimates are derived from associated metamafic rocks and metasediments while for metaserpentinites these are semi-quantitative and primarily constrained by temperature-dependent phase transitions, in the modified manuscript (lines 106-111): “It must be noted that P-T constraints for metaserpentinites remain inherently uncertain due to the limited applicability of classical geothermobarometers in ultramafic rocks. In this study, P–T estimates for metaserpentinites are therefore semi-quantitative and are primarily constrained by temperature-dependent phase transitions, while pressure estimates rely on associated lithologies (metabasalts, matagabbros and metasediments; see Supplementary Material Figure S1).”

We have expanded the methodological description in the Supplementary Material (Caption of Fig. S1c), where we now explicitly describe the phase transitions used as constraints, including the lizardite–antigorite transition (300–400 °C), brucite breakdown (450–550 °C) and the titanoclinohumite–titanochondrodite transition (2.2–2.5 GPa) that have been used for semi-quantification on PT record in metaserpentinites. In addition, Figure S1c has been revised to illustrate these phase transitions. We believe these clarifications strengthen the robustness and transparency of the P–T framework.

Additionally, I encourage the authors to compare their organic carbon data with metamorphic graphite produced by the reduction of sedimentary carbonates by serpentinite-derived fluids in the Alps. This comparison could further contextualize the isotopic signatures and deepen the discussion regarding the origins of light ^{13}C values.

We thank the reviewer for this helpful suggestion. We have now added a direct comparison between the $\delta^{13}\text{C}$ signatures of organic carbon in metaserpentinites and those reported for metamorphic graphite formed by reduction of sedimentary carbonates in the Western Alps (citing the studies of Galvez et al., 2013 and Vitale-Brovarone et al., 2017, see lines 320-325 of the modified manuscript). This comparison highlights the distinct isotopic compositions of these two processes and further supports our interpretation that the light $\delta^{13}\text{C}$ values preserved in metaserpentinites are not consistent with carbonate reduction during subduction.

Below, I list minor comments aimed at further improving clarity and completeness:

Minor comments

1. Lines 65–66: In addition to the authors' point, it may be worth noting that sediments are often not recycled into the sub-arc mantle due to their relatively low density, which can promote diapiric ascent instead of continued subduction.

We thank the reviewer for this suggestion. We have added a sentence noting that sediments may detach from the downgoing slab to form buoyant diapirs, potentially limiting their transfer to the sub-arc mantle and deep recycling: "Furthermore, subducted sediments may detach from the downgoing slab to form buoyant diapirs and therefore escape from a deep recycling"¹⁶

2. Line 74: The term *en route* is French. For stylistic consistency, consider replacing it with an English equivalent.

We rephrased as following: "These represent a previously unexplored pathway for organic carbon recycling into the deep mantle in subduction settings."

3. Line 94: The P–T range of 415–475 °C / 1.7–2.2 GPa corresponds to eclogite-facies conditions. Please clarify this categorization or adjust the phrasing.

In the manuscript, we follow the established regional terminology used for the Western Alps, without attempting to redefine metamorphic facies, as this is not the focus of the present study. The metamorphic facies in the Western Alps are defined by the presence of mineral assemblages rather than P–T conditions. In this specific unit at 415–475 °C / 1.7–2.2 GPa the mineral assemblage in mafic rocks are still highly hydrated, typical of blueschist facies (Gln-Lws-±Jd, No garnet). This contrast with the eclogite facies units, which contain classical eclogite facies assemblages (Gt-Omp-Qz/Co). To avoid any confusion with literature we rephrase as following: "which preserves higher-temperature blueschist transitional to eclogite facies conditions (415–475 °C / 1.7–2.2 GPa)"

4. Lines 133–139: Please provide appropriate references for the spectral peak assignments.

This as been specified in the figure caption; the assignment is from Pasini et al. (2013) and Stuart (2004).

5. Lines 149–155: References are also required here. Alternatively, a supplementary table summarizing all peak assignments would improve clarity.

The peak assignments were specified in the figure caption.

6. Lines 273–274: The conclusion that the degree of disorder in abiotic organic compounds is not P–T dependent (based on PAH simulations) may require additional high-pressure experimental evidence. Please clarify or temper this statement.

We thank the reviewer for the comment. PAH are considered to represent the best analogues of abiotic solid organic compounds at abyssal stage. Recent studies have already been using these compounds for this purpose in thermodynamic calculations at abyssal stage (Milesi et al. 2016; Andreani et al. 2023). This is specified line 304–305 of the modified manuscript.

7. Line 285: Further explanation is needed regarding how the authors confirm that the metaserpentinites reached eclogite-facies conditions, given the well-known difficulty of constraining P–T in meta-ultramafic rocks.

See our response to main comment.

8. Line 339: Could the generally low temperatures (<300–400 °C) documented for most metaserpentinites be the primary factor driving their low ^{13}C signatures? A brief discussion would strengthen the interpretation.

While some metaserpentinites may preserve low-temperature assemblages, the studied samples record a broad P–T range (from ~300–400 °C to 600°C), as indicated by their mineralogical assemblages. Therefore, the generally low $\delta^{13}\text{C}$ values are not solely driven by low temperatures but rather reflect the preservation of abyssal organic carbon during prograde metamorphism. See response to main comment.

Reviewer #2 (Remarks to the Author): Comments to manuscript by Debret et al:

Debret et al., measured the inorganic and organic carbon contents and carbon isotope ratios, together with nitrogen contents, of both meta-sediments and meta-serpentinites with different metamorphic grades, aiming to understand (and emphasize) the role of meta-serpentinites in transferring isotopically light ^{12}C to the mantle, contributing to the diamond-forming fluids with low $\delta^{13}\text{C}$ values (e.g., eclogitic diamonds). They observed the equilibration of carbon isotopes between inorganic carbonate and organic carbon in meta-serpentinites during prograde metamorphism, thus tending to homogenize the $\delta^{13}\text{C}$ values of the sedimentary component. This is however not observed in the adjacent meta-serpentinites, highlighting the role of meta-serpentinites in transporting ^{12}C -enriched organic carbon to the mantle.

Overall, I agree with the proposed idea of meta-serpentinites in transferring ^{12}C -enriched carbon to the mantle, but their relative importance compared to the seafloor sediments and altered oceanic crust needs to be further evaluated. Given the overall quality of the data and the logic that authors applied to reach conclusions, I would recommend major revision after these two main points and the following minor points were reworked.

One of the major comments is: the authors stated that the $\delta^{13}\text{C}$ values of the bulk $\delta^{13}\text{C}$ value of subducting sediments after re-equilibration during prograde metamorphism are comparable to that of the general mantle, thus concluding that sedimentary components cannot transport carbon with low $\delta^{13}\text{C}$ values. This needs to be evaluated, since sedimentary carbonate will also be mobilized out of the slab and the remaining carbonate may not significantly modify the $\delta^{13}\text{C}$ values of the organic carbon. Though this is not observed in your samples, but this loss process of carbonate via decarbonation/carbonate dissolution (etc.) needs to be really discussed, since there are numerous studies discussing the carbonate mobilization from the slab to the forearc and arc volcanoes in modern subduction zones. The authors calculated the bulk $\delta^{13}\text{C}$ values of the subducting sediments of modern subduction zones, so these processes need to be evaluated and is important to your following discussions of the role of meta-serpentinites in deep carbon cycling. If you consider these processes, would the sedimentary component play a role in transferring ^{12}C -enriched organic carbon to the mantle? You may leverage the findings from Farsang et al., 2021 (Nature Communications, Deep carbon cycle constrained by carbonate solubility), where the mobilization of carbonate from slab at different depths have been evaluated (see my detailed comments below). This will need some work in the mass balance calculations of each individual subduction zones to calculate the final $\delta^{13}\text{C}$ values of sedimentary carbonate after some carbonate is lost.

We thank the reviewer for highlighting this important point. We have now explicitly evaluated the effect of carbonate loss during subduction on both the isotopic composition and the mass balance of recycled sedimentary carbon.

First, we explored the isotopic consequences of carbonate devolatilization using a first-order Rayleigh distillation model based on CO_2 -carbonate equilibrium fractionation ($\Delta\text{CO}_2\text{-calcite} \approx 3\text{‰}$ at 600–700 °C; Chacko et al., 1991). The results, presented in the new Supplementary Fig. S4, show that isotopic fractionation remains limited unless carbonate loss is nearly complete ($F < 0.1$).

We then integrated slab-specific carbonate recycling efficiencies from Farsang et al. (2021), which account for both thermal structure and water availability, into this Rayleigh framework. This approach allows us to evaluate the combined effect of carbonate loss and isotopic fractionation for individual subduction zones. The resulting predictions are presented in the revised Fig. 5b.

These calculations indicate that while carbonate loss has only a minor effect on the $\delta^{13}\text{C}$ signature of the recycled sedimentary carbon (maximum shift of 5 ‰ in carbonate rich settings while insignificant in organic rich systems i.e., $\ll 1$ ‰), it strongly reduces the efficiency of carbon transfer from metasediments to the mantle at the global scale. As a consequence, even if some organic carbon remains isotopically light, the overall contribution of sediments to the deep mantle carbon budget is substantially diminished once carbonate mobilization is considered. Importantly, incorporating these processes further supports our main conclusion that metaserpentinites represent a distinct and efficient pathway for recycling isotopically light organic carbon, rather than weakening it.

The revised manuscript now includes a section discussing the impact of decarbonation or carbonate dissolution (new Fig5b and lines 267-277 of the revised manuscript: “In addition to isotope re-equilibration, carbonate loss through decarbonation or carbonate dissolution may further affect both the carbon budget and isotopic composition of subducting sediments. Recent work by Farsang et al. (2021) quantified slab-specific carbonate recycling efficiencies as a function of thermal structure and water availability. To evaluate the isotopic consequences of these processes, we implemented a first-order Rayleigh distillation model using a CO_2 –calcite fractionation factor of $\sim 3\%$ at 700 °C (see Supplementary Methods). The results (Fig. 5b) indicate that carbonate loss has a limited impact on the $\delta^{13}\text{C}$ signature of recycled sedimentary carbon, unless carbonate removal is nearly complete. In contrast, the efficiency of sedimentary carbon recycling decreases sharply with increasing carbonate loss. Together these findings imply that carbonate mobilization primarily affects the magnitude of carbon transfer rather than its isotopic signature.”) and we present the details of these calculations in supplementary materials (revised Tables S6 & new Fig. S4).

Another major comments: when the authors are trying to emphasize the role of meta-serpentinites in transporting organic carbon to subduction zones, the authors compared the global carbon flux of meta-serpentinites with some individual subduction zones, showing that this global carbon flux of meta-serpentinites are comparable to some subduction zones with sedimentary carbonate-rich lithologies, and a hundred times higher than the subduction zones dominated by sedimentary organic carbon. This does not sound to compare a global flux with an individual subduction zone. For your purpose, the authors should compare unit flux in Mt C/yr/Km to better get a sense of their relative importance. If you do this unit flux comparison, is the meta-serpentinites still important in transporting isotopically depleted carbon to the mantle?

We thank the reviewer for this important and constructive suggestion. We have revised our approach and now compare carbon fluxes normalized to subduction length (T/yr/km), allowing a more meaningful comparison between serpentinite subduction and individual sediment-dominated systems.

Using this framework, we find that the unit carbon flux associated with serpentinitized peridotites (~ 255 T/yr/km) is comparable to, or up to an order of magnitude higher than, that of organic-rich sedimentary systems such as Izu-Bonin, Kuril, and NE Japan, while remaining lower than carbonate-rich subduction zones (see revised Table S6).

*We have accordingly revised the manuscript to restrict comparisons to organic-rich systems and removed statements implying that serpentinites represent the most efficient carbon recycling pathway. These changes do not affect our conclusions regarding the distinct $\delta^{13}\text{C}$ signature of serpentinite-derived carbon or its potential role in deep N mantle recycling. The revised comparison is presented in the main text lines 384-397 and in the updated Table S6: “When normalized to subduction length (**Table S6**), the carbon influx from serpentinitized peridotites (~ 255 T/yr/km) is comparable to, or up to an order of magnitude higher than, that of sediments in organic-rich subduction systems such as Izu-Bonin (42 T/yr/km), Kuril (148 T/yr/km), and NE Japan (410 T/yr/km). While lower than carbonate-rich systems,*

these fluxes indicate that serpentinite subduction represents an efficient and previously underappreciated pathway for the recycling of isotopically light organic carbon into the deep mantle.”

Other comments-Keyed to line numbers:

L 55-56: references for $\delta^{13}\text{C}$ values of organic carbon and carbonate in subducting sediments

We added Cartigny et al. (2014).

L 55: organic carbon is isotopically light

Modified accordingly

L 126: the mineral assemblage of meta-serpentinite is listed in Table S1.

*Modified as : “The mineral assemblages of metaserpentinite are listed in **Table S1.**”*

L 191: lighter

Modified accordingly

L 223: poorly sensitive to insensitive

Modified accordingly

L 231-233: Are there any systematic correlations between TIC or TOC contents with metamorphic grades? This could be useful for your argument here to argue against significant decarbonation.

The [CTIC] and [CTOC] concentrations of abyssal sediments are highly variable (over 3-4 orders of magnitude) and fully overlap with HP metasediments (see Fig. S3). The use of [CTIC] and [CTOC] concentrations as a reliable proxy for C mobility during sediment dehydration in HP-HT samples will require to 1- trace the protolith influence (i.e., deciphering between organic and carbonate rich protoliths using geochemical proxies such as CaO and CaO+SiO₂+K₂O) and 2- decipher between protolith variations and potential loss during prograde metamorphism (combination of C concentrations, isotopes and major element proxies). Such type approach was already attempted in the Western Alps (see for example Busigny et al. (2003, 2011) or Qu et al. (2023)) who were not able to evidence a significant C mobilization.

We specify that we were specifically talking about Western Alps samples and modified our statement as following: “Such a nearly constant values support only modest decarbonation in alpine metasediments during subduction, as already highlighted by previous isotopic studies^{12,38} and flux estimates⁹”

L 234: the average C isotope fractionation between carbonates and solid organic compounds..... Please also report the values of this $\Delta^{13}\text{C}$ based on your samples.

The average values are compared with our sample analyses in Fig. 4c. We delete the word average to avoid any confusion.

L 252: Please specify here clearer: what is the meaning of full re-equilibration? Do you mean re-equilibration at a certain temperature between inorganic and organic carbon? Or this re-equilibration temperature does not matter, since organic carbon and inorganic carbon will reach equilibrium finally via isotope exchange their overall $\delta^{13}\text{C}$ values will remain unchanged if it is a close system without carbon loss. This is important to your calculation of re-equilibrated $\delta^{13}\text{C}$ values of the subducting meta-sediments at global scale, so this needs to be clearly specified.

The Δ should reach ~ 0 between 600 and 800°C according to empirical fitting of organic and inorganic carbon re-equilibration, with temperature based on natural data in subduction zones (trends H21 and this study in the Fig. 4c). The exact temperature is not significant.

We specified during prograde metamorphism “Considering a full re-equilibration during prograde metamorphism between organic and inorganic carbon at the slab scale”

L 250-252: The ratio of CTOC/CTC you defined here is the flux of organic carbon divided by total carbon flux (organic + inorganic), right? Please make it clear.

Yes this is based on Clift (2017), this is now specified as following: (C_{TOC}/C_{TC} based on flux estimates of organic and total carbon by ¹⁰)

L 516: Pay attention to your mass balance calculation equation: the second item should be (1-CTOC/CTC)* $\delta^{13}C_{CTC}$? I believe that your calculation results are right (checked), but you just put the bracket in wrong position.

Sorry for the typo, it has been modified accordingly.

L 262: remove cratons, since kimberlites erupted on cratons.

Modified accordingly

L 250-270: I agree with your arguments that isotope re-equilibration between organic and inorganic carbon tends to move the $\delta^{13}C$ value of the entire subducting sedimentary component to higher values and thus erase the $\delta^{13}C$ values of primary organic carbon with low $\delta^{13}C$ values of -22 ‰ via equilibration. The final $\delta^{13}C$ values of the subducting sedimentary component is dependent on the budget of organic vs. inorganic carbon. What if some sedimentary carbonate is removed from the sediments via forearc or arc volcanoes, via decarbonation and carbonate dissolution? Your study is based on Alps samples, and it seems minor decarbonation processes happen, but these effects have been widely observed and modelled in subduction zone settings and are important pathway for carbon removal from subducting slab (see studies on geochemistry of volcanic gases). If organic carbon has remained in the slab as refractory graphite, while some carbonate is removed from the subducting slab. Will this change the overall $\delta^{13}C$ value of the subducting sediments that are transferred to the mantle? This might not be easy to quantify from a global perspective, but studies from Central America and Izu-Bonin-Mariana have numerous datasets on the fluxes of volcanic CO₂ that is ultimately sourced from the subducting sediments. This needs some discussion and evaluation on your calculation of final $\delta^{13}C$ values of the subducting sediments to the mantle from a global perspective.

We evaluate this as requested in the main comments and provide full detailed calculation – see answer to main comment.

L 275-276: do you mean that these organic compounds were produced and trapped during the fluid-rock interactions inside the subduction zones? In L 281-282: the authors stated that the organic compounds found in the meta-serpentinite are similar to those found in the serpentinized peridotites in abyssal settings. Is it possible that these organic compounds found in meta-serpentinites of this study come from the seafloor serpentinization processes?

We mean that they are trapped during the growth of HP-HT minerals. We then compare the structure of these organic compounds with maturation experimental works suggesting that they can be inherited from an abyssal stage. We further reinforced this argumentation by stating that the structure of these compounds differs significantly from those formed in open-system HP conditions, such as in forearc

serpentinites, where organics are formed by FTT type reaction and dominated by a strong aliphatic structure.

To clarify this we specified: “The observation of solid organic carbon in HP-HT metamorphic phases (Fig. 2a-b & Fig. 3, see also ³²) suggest that these were trapped during the growth of prograde metamorphic minerals.”

L 295-297, L 197-198: note that in L 197-198, the authors stated that the $\delta^{13}\text{C}$ values of organic carbon and inorganic carbon is larger in meta serpentinites (-36 to -28‰; -12 to -2‰), but in L 295-297, the authors stated that $\delta^{13}\text{C}$ of organic carbon remains constantly (-36 to -22‰), while $\delta^{13}\text{C}$ of inorganic carbon show little variation (-8 to 0‰). The reported $\delta^{13}\text{C}$ values are not consistent between L 197-198 and L 295-297. This needs to be clarified.

Very sorry there was indeed a typo lines 295-297, it is from -36 to -28 ‰ and from -12 to -2‰, thanks for spotting this.

Table S5: add the lithologies for each of the sample you have analyzed. It's not very convenient to look at the ID in table S5 and find the corresponding lithology in table s1.

Yes, you are right, actually we think that the previous typo was a consequence of this. We specified M: metasediments, S: metaserpentinite in Table S1 to help the reading of the dataset.

L 304-305: report your C_{TOC}/N ratio of your meta-serpentinite samples here.

Modified accordingly: Living bacteria have an atomic C_{TOC}/N of 10 and negative $\delta^{13}\text{C}_{\text{TOC}}$ (~ -25 ‰ ³⁶). In contrast, the metaserpentinites investigated here exhibit a wide range of C_{TOC}/N ratios, from 7 to 46.

L 308-309: is there any correlation between fluid-mobile elements (e.g., Cs, As and B) and TOC or N contents in these meta-serpentinites?

We could not find any correlation with other fluid-mobile proxies, see for example:

L 314-316 and L 332-333: In L 314-316, the authors stated that the organic carbon was hosted in metamorphic products, such as antigorite, chlorite and olivine, which was formed during early stage

of subduction. In L 332-333, the authors stated that the organic carbon was likely sourced from oceanic processes. I am not very clear about these two statements. Do the authors try to argue for/against the origin of organic compounds from serpentinization in abyssal settings or inside the subduction zones? From fluid-mobile elements, it is clear to me that the authors are trying to argue for the production of organic compounds during the oceanic serpentinization processes? This needs to be clarified here.

We thank the reviewer for pointing out this ambiguity. We clarify that we argue for an oceanic (abyssal) origin of the organic compounds, formed during serpentinization. These compounds were subsequently preserved and trapped within metamorphic minerals (e.g., antigorite, chlorite, olivine) during prograde metamorphism in subduction zones.

To avoid any confusion, we have revised the manuscript to explicitly distinguish between the formation environment of the organic compounds and their metamorphic mode of preservation: "Overall, both petrographic and geochemical observations indicate that solid organic compounds formed during seafloor serpentinization (abyssal setting) and were subsequently preserved by encapsulation within metamorphic minerals during prograde subduction-related metamorphism."

334-344: carbonate precipitated in oceanic lithosphere could be highly variable and this could be caused by the relative contributions of inorganic carbonate from abiotic processes (general precipitation of bicarbonate from hydrothermal fluids) and microbial processes (see Furnes et al., 2001 and series papers published by him and his colleagues). Microbial fractionation of carbon isotopes in altered basaltic glass from the Atlantic Ocean, Lau Basin and Costa Rica Rift). Then, the obvious question is would you expect that these $\delta^{13}\text{CTIC}$ are a result of microbial processes?

We don't have any micro- to nano-observation on the carbonate. So, we can only speculate. Microbial fractionation of carbon isotopes in carbonates are usually highlighted through authigenic compositions (i.e., highly negative $\delta^{13}\text{CTIC}$), as we don't have neither evidence for biological activity when looking at the organic carbon structure or isotopic signature, we did not discuss in details such a process.

L 336-338: The highest disequilibrium occurs in the samples with highest metamorphic grade based on Fig. 4c. If it is the leaching (dissolution) of inorganic carbonate during subduction processes, do you see the loss of inorganic carbon during the progressive subduction of the meta-serpentinites based on your samples, particularly those with highest $\Delta^{13}\text{CTIC-TOC}$, likely representing the most obvious disequilibrium caused by carbonate leaching.

We thank the reviewer for this comment. In our dataset, we do not observe clear evidence for CO_2 leaching during progressive subduction of the metaserpentinites. Both carbon concentrations and $\delta^{13}\text{CTIC}$ values remain largely consistent with those of abyssal serpentinites (Fig. S3c), even in samples exhibiting the highest $\Delta^{13}\text{CTIC-TOC}$. We can only speculate on the significance of the $\delta^{13}\text{CTIC}$.

L 345-358: Here, the authors stated that meta-serpentinites could be an important reservoir contributing organic carbon to the mantle sources, which I agree. But they stated that carbon fluxes of global serpentinites are comparable to some subduction zones with carbonate-rich lithologies and extremely higher than the subduction zones dominated by organic-rich sediments. This comparison is not clear to me. It does not sound if you compare a global total flux of carbon carried by serpentinites with a specific subduction zone at a local scale. What if this carbonate-rich/organic-rich sediments dominate the subduction zones at global scale? Then, does these serpentinites still play an important role with comparable carbon fluxes to the sediments? If you really want to emphasize the role of meta-serpentinites in transferring carbon to subduction zones, it is better to compare them with a unit flux, i.e., Mt C/yr/km. This will give a more straightforward to compare

between carbon fluxes from meta-serpentinites and sediments. E.g., you can calculate the unit flux of meta-serpentinites from a global perspective (the flux has been estimated by early studies) and compare this unit flux with individual subduction zones (e.g., carbonate-rich versus organic carbon rich). Based on this kind of comparison, do you still see that unit flux of meta-serpentinites is comparable or higher than the sediments carbon flux? Another thought is that: if you want to highlight the subduction or isotopically light carbon to the mantle via subduction of meta-serpentinites, you may simply compare the meta-serpentinites with subduction zones dominated by the subduction or organic carbon, since these subductions are expected to be the main carrier of ^{13}C -depleted organic carbon, compared to the other subduction zones dominated by isotopically heavy carbonate.

We fully agree with this comment – we modified accordingly the manuscript, see answer to main comment.

L 382-384: when you calculate the flux of nitrogen using C_{TOC}/N, what is the flux of organic carbon did you use for the calculation? This needs to be clearly stated.

Sorry about this, the organic/carbonate ratio is considered of 20% based on Plank & Manning (2019). The total flux of carbon is variable in the literature, we therefore considered extreme ranges. This is modified as following: “Considering a constant C_{TOC}/N of 10 and an organic flux of 8 to 12 MtC/yr (considering 20% of organic carbon⁹ and total carbon fluxes from ^{8,10}), we re-evaluate the amount of subducted N by metasediments to 0.8 to 1.2 MtN/yr.”

L 396-397: there is no detectable N-bounds in chlorite. Did you also test the serpentinite minerals? Do they show any signature of N bonding?

There are neither N bonding in serpentine minerals see Fig. 2. This is now highlighted in the text: “phyllosilicate, such as chlorite (or antigorite), barely display detectable N-bounds (Fig. 2c & Fig. 3c)”

L 410: previous estimation of N flux by seafloor serpentinites are within the range of your estimation of N flux, but your estimation goes to higher values. Make this cleaner.

Modified accordingly, as following: “Although these estimates remain subject to revision, as the serpentinite reservoir in subduction zones is still poorly constrained, our results overlap with previous nitrogen isotope-based estimates ($\sim 0.1 \text{ Mt N yr}^{-1}$)⁶¹, but suggest that nitrogen fluxes from serpentinites may reach higher values.”

Other minor comments worth considering:

This manuscript basically compared the relative importance of carbon in meta-serpentinites versus the sedimentary carbon. There are numerous works emphasizing the role of carbon transportation to the subduction zones by altered oceanic crust (AOC). I think it is at least worth some discussion when you emphasize the role of meta-serpentinites in the deep carbon recycling, instead of stating that the carbon fluxes of subduction zones are dominated by sedimentary reservoir and AOC-bearing carbon is not important or negligible (see Alt et al., 1999, GCA, The uptake of carbon during alteration of ocean crust; Coogan and Gills, 2011, Secular variation in carbon uptake into the ocean crust). Instead, in the subduction zones where sedimentary carbon is dominated by organic carbon, AOC carbonate could dominate the budget of subducting slab (such as Izu-Bonin, Tonga, and Kurile; see Farsang et al., 2021; Nature Communication, Deep carbon cycle constrained by carbonate solubility), and in particular, the carbon isotope compositions of carbonate in oceanic crust is highly variable and can also contribute carbon isotopes with extremely low- $\delta^{13}\text{C}$ values (see Furnes et al.,

2001; Li et al., 2019). I understand that a global perspective of $\delta^{13}\text{C}$ values of AOC is still loosely constrained, but this needs to be mentioned instead of just being neglected.

We thank the reviewer for highlighting the potential role of altered oceanic crust (AOC) in carbon transfer to subduction zones. We agree that AOC can represent a significant carbon reservoir, including organic carbon acquired during abyssal alteration, as previously suggested (e.g., Shilobreeva et al., 2011). We have now explicitly acknowledged this in the manuscript (lines 403-407). However, the evolution of organic carbon in AOC during prograde subduction metamorphism remains poorly constrained, which currently limits its quantitative integration into global deep carbon cycling models. Our study therefore focuses on serpentinites, for which we provide direct constraints on organic carbon preservation and isotopic evolution.

The authors analyzed the carbon content and isotope compositions, as well as nitrogen contents of meta-sediments and meta-serpentinites by isotope ratio mass spectrometers. I wonder why the nitrogen isotope compositions were not measured, but just nitrogen contents?

We thank the reviewer for this comment. Nitrogen isotope measurements in these samples are technically challenging due to the low nitrogen contents, which are close to the detection limits of conventional IRMS techniques. In the present study, we therefore focused on nitrogen concentrations, which already provide robust constraints on the origin and preservation of organic nitrogen and are sufficient to address the questions investigated here.

We note that the development of combined analytical approaches for low-level nitrogen isotope measurements is currently ongoing through collaborative work, but these developments fall beyond the scope of the present manuscript and will be addressed in future studies.

Reviewer #2 (Remarks to the Author):

I have now thoroughly reviewed the revised manuscript. My suggestions have been worked in and the manuscript presents itself significantly improved over last version.

I only have one comment need to be cleared:

Regarding the calculation of isotope fractionation during carbon removal from subducting slab, if I am understanding the model right, the authors considered the decarbonation mechanism for carbon removal as CO₂. How about dissolution of carbonate as bicarbonate in fluids? This will result in different isotope fractionation factors between Calcite-CO₂ and Calcite-HCO₃⁻ (in aqueous fluids) The authors could use carbonate dissolution as another extreme endmember of carbon removal to evaluate its effect on δ¹³C values of the remaining carbon in slab. After this comment is addressed, I can recommend the revised manuscript for publication in Nature Communications.

We thank the reviewer for this insightful comment. At high temperatures relevant to subduction-zone conditions, isotope fractionation factors between calcite and aqueous carbon species such as HCO₃⁻ or CO₃²⁻ remain poorly constrained. Existing estimates are largely based on extrapolations from low-temperature experimental data (e.g., Chacko et al., 2001 – see figure below).

Importantly, these extrapolations suggest that calcite– HCO₃⁻ aqueous species fractionation factors are negative, resulting in fluids that are isotopically lighter than the residual carbonate. Therefore, modelling carbon removal exclusively as CO₂ represents an endmember scenario that maximizes the isotopic shift toward lighter residual carbon in the slab. Inclusion of aqueous carbonate species would act in the opposite direction and reduce or counterbalance this effect.

To clarify this point, we have added the following sentence to the manuscript: “If present, other aqueous fluid species (e.g., HCO₃⁻ and CO₃²⁻) mixed with CO₂ in slab-derived fluids are expected to reduce this fractionation factor, as they exhibit negative fractionation relative to calcite⁴⁵; the modelled fractionation therefore represents a maximum endmember.”

Figure 20 from Chacko et al. (2001) displaying fractionation factors between various geological material and CO₂. Note that at high temperature, in aqueous fluids, only CO₂ is expected to be isotopically heavy relative to calcite.